# Enhanced Baseflow Separation in Rural Catchments: Event-Specific Calibration of Recursive Digital Filters with Tracer-Derived Data

Fernanda Helfer[1], Felipe Bernardi[2], Claudia Alessandra Peixoto de Barros[3], Daniel Gustavo Allasia[2], Jean Paolo Gomes Minella[4], Rutinéia Tassi[2], Néverton Scariot[2]

[1]School of Engineering and Built Environment, Griffith University, Gold Coast, QLD, 4222, Australia
[2]Sanitary and Environmental Engineering Department, Federal University of Santa Maria, Santa Maria, RS, 97105-900, Brazil.
[3]Soil Department, Federal University of Rio Grande do Sul, Porto Alegre, RS, 91540-000, Brazil.
[4]Soil Department, Federal University of Santa Maria. Santa Maria, RS, 97105-900, Brazil.

*Correspondence to*: Fernanda Helfer (f.helfer@griffith.edu.au)

**Abstract:** This study investigates the performance of baseflow separation methods in a small rural catchment, emphasizing the calibration of three Recursive Digital Filters (RDFs): Eckhardt, Lyne and Hollick (LH), and Chapman and Maxwell (CM). We introduced silica concentration as a reference tracer, and refined the parameterization of $BFI_{max}$ in the Eckhardt's filter and *Beta* in the LH filter. An innovative event-specific calibration methodology was applied, where rainfall events were categorized by intensity to tailor filter parameters accordingly. Results indicate that the Eckhardt's filter, when calibrated dynamically per event magnitude, yields the most accurate baseflow estimates, closely aligning with observed data. The event-based calibration significantly enhanced accuracy, particularly for the Eckhardt's and LH filters, compared to a general calibration method. The CM filter, despite generating reasonable hydrograph shapes, consistently underestimated baseflow due to its fixed parameters. These findings highlight the necessity of customized calibration strategies for improved baseflow separation and underscore the superior performance of the Eckhardt's filter when integrated with event-specific calibrations.

## 1. Introduction

Baseflow, a key hydrological component sustaining river flow during dry periods, originates from groundwater discharge and delayed subsurface contributions. Accurate quantification of baseflow is essential for managing water resources, designing infrastructure, and maintaining ecological stability (Apurv & Cai, 2020; Beatty et al., 2010; Glas et al., 2023; Gómez et al., 2020; Miller et al., 2016; Murray et al., 2003; Walker, 2023). However, direct measurement remains challenging due to complex interactions between surface and subsurface hydrological processes.

Measuring baseflow presents significant challenges, often requiring extensive resources due to the reliance on tracer-based methodologies or aquifer-level monitoring, both of which can be costly and time-intensive (Vasconcelos et al., 2013). To address these limitations, Recursive Digital Filters (RDFs) have been developed as an alternative approach for estimating baseflow using streamflow data (Chapman & Maxwell, 1996; Eckhardt, 2005; Furey & Gupta, 2001; Lyne & Hollick, 1979). Although most RDFs do not explicitly simulate the underlying physical mechanisms governing baseflow – except for the model proposed by Furey & Gupta (2001) – they remain widely utilized in hydrological studies. These methods are particularly valued for their ability to produce hydrologically plausible baseflow hydrographs with smooth transitions (Xie et al., 2020), generate consistent and reproducible results (Arnold et al., 1995; Ladson et al., 2013; Su et al., 2016), and offer computational efficiency, making them easy to implement and automate (Li et al., 2013; Su et al., 2016).

In addition to streamflow data, RDFs require specific parameters that are designed to accentuate the slowly changing, persistent characteristics associated with baseflow hydrographs, while attenuating the faster fluctuations related to quickflow (Chapman & Maxwell, 1996; Eckhardt, 2005, 2008; Lyne & Hollick, 1979; Nathan & Mcmahon, 1990; Sloto & Crouse, 1996). Therefore, choosing the right parameters is crucial for RDFs to produce reliable outcomes. There is a growing trend towards utilizing independent baseflow estimates (such as tracer-based baseflow) in RDF calibration (Gonzales et al., 2009; Helfer et al., 2024), even though the filters offer non-calibration opportunity, with the selection or derivation of filter parameters based on catchment characteristics (Chapman & Maxwell, 1996; Eckhardt, 2005; Nathan & Mcmahon, 1990).

The simplest RDF filters are those that rely on one single parameter. Two widely used one-parameter filters are the models developed by Lyne and Hollick (1979), referred to as 'LH', and Chapman and Maxwell (1996), referred to as 'CM'. The LH model is expressed as:

$$qfLH_t = \beta \times qfLH_{t-1} + \frac{1+\beta}{2} \times (q_t - q_{t-1}) \quad , \quad \text{subject to } qf_t \geq 0 \tag{1}$$

where $qfLH_t$ is the quickflow at time $t$, $qfLH_{t-1}$ is the quickflow at time $t$-$1$, $q_t$ is the total streamflow at time $t$, and $q_{t-1}$ is the total streamflow at time $t$-$1$. The filter parameter, $\beta$, regulates the separation process. Baseflow at each timestep, $qbLH_t$, is determined by subtracting quickflow from the total streamflow, following the equation $qbLH_t = q_t - qfLH_t$. This filtering process is typically executed in three iterations: a forward pass, a backward pass (where the $t$-$1$ timestep is replaced by $t$+$1$), and a final forward pass. During the first forward pass, $q_t$ is substituted with the baseflow calculated in the preceding pass.

These sequential iterations help to refine the baseflow hydrograph by smoothing fluctuations and introducing a slight delay in the baseflow peak relative to quickflow (Murphy et al., 2011).

For the *Beta* parameter ($\beta$), the common reference value is 0.925 (Arnold & Allen, 1999; Arnold et al., 1995; Ladson et al., 2013; Nathan & Mcmahon, 1990), with values within the range 0.90 – 0.95 claimed to yield the most reliable baseflow hydrographs (Nathan & Mcmahon, 1990). However, comparisons of modelled and measured baseflow values derived from tracer observations have indicated optimal values above this range. For example, a study in the Murray Darling Basin by CSIRO & SKM (2010) reported by Ladson et al. (2013) found a *Beta* value of 0.98 to yield more accurate baseflow separations. This finding is supported by Zhang et al. (2017), who identified *Beta* values between 0.943 to 0.987 across five Australian catchments as more effective. Li et al. (2013) noted the limitations of the 0.925 *Beta* parameter, especially in catchments with soils of low hydraulic conductivity, pointing out the necessity for catchment-specific calibrations due to the filter's sensitivity. Conversely, Mau and Winter (1997) identified a *Beta* value of 0.85 as optimal for a small mountain catchment in New Hampshire, noting that values above 0.85 led to unrealistic baseflow estimates. The recommendations to utilize the recession constant as the *Beta* parameter by researchers such as Eckhardt (2005), Tan et al. (2009a), Zhang et al. (2017), Mugo and Sharma (1999) accentuate the variability in selecting appropriate *Beta* values across hydrological studies. This variability emphasizes the need for further investigation and catchment-specific adjustments, as reinforced by Ladson et al. (2013). This calibration becomes even more important due to the high sensitivity of the filter to the *Beta* parameter. For instance, Nathan and Mcmahon (1990) observed that a ±3% change in the parameter could alter the baseflow index (*BFI* – the long-term ratio of baseflow volume to the total volume of streamflow) value by as much as +14% and -26%, highlighting the critical nature of precise *Beta* parameter selection.

Despite these challenges in parameter determination, the LH method has been extensively applied in hydrology. It is the approach used in the "*Low flow atlas for Victorian streams*" (Nathan & Weinmann, 1993) and the method recommended by the "*Australian Rainfall and Runoff*" guidelines (Murphy et al., 2011). It is also the algorithm used in the BFLOW program (Arnold & Allen, 1999). Other examples include the work by Lacey and Grayson (1998), who applied the method in 114 catchments with areas < 192 km$^2$ in Victoria (Australia), the work by Nathan and McMahon (1992), who applied the method to 186 catchments in Victoria and New South Wales (Australia), research by Mugo and Sharma (1999) who applied the filter in three catchments in Kenya with areas ranging from 0.36 to 0.65 km$^2$, and the studies by Tan et al. (2009a; 2009b), who applied the method to a 5.6 km$^2$ in Singapore.

The CM model (Chapman & Maxwell, 1996) is another one-parameter filter and is given as:

$$qbCM_t = \frac{a}{2-a} \times qbCM_{t-1} + \frac{1-a}{2-a} \times q_t \qquad (2)$$

where $qbCM_t$ is the CM filter baseflow at time $t$, $qbCM_{t-1}$ is the CM filter baseflow at time $t-1$, and $a$ is the filter parameter, given by the baseflow recession constant. In the CM and LH models, when the calculated baseflow at a specific time step exceeds the total streamflow, the baseflow value is capped at the total streamflow value.

The CM method has been largely studied and applied in hydrology (Chapman & Maxwell, 1996; Cheng et al., 2022; Indarto et al., 2016; Kang et al., 2022; Lei et al., 2011; Qiutan & Yong, 2019; Tan et al., 2009a; Tan et al., 2020). A key advantage of this method is its reliance on a single parameter – the recession constant (assumed to be time-invariant), which simplifies the employment of this filter. This constant can be readily and reliably determined by analyzing the recession sections of observed streamflow hydrographs (e.g. Tallaksen, 1995). Notably, research has shown that better separation typically occurs when the recession constant value is high (> 0.90) (Indarto et al., 2016; Kang et al., 2022).

Another largely employed RDFs is the Eckhardt's filter (Eckhardt, 2005), a two-parameter model, given as:

$$qbECK_t = \frac{(1 - BFI_{max}) \times a \times qbECK_{t-1} + (1 - a) \times BFI_{max} \times q_t}{1 - a \times BFI_{max}} \qquad (3)$$

where $BFI_{max}$ [0,1] is the model parameter (time-invariant), representing the highest $BFI$ that the filter can compute.

The Eckhardt's filter is usually preferred over other RDFs for more accurate baseflow values and versatility across catchment sizes and types, as demonstrated by various studies (Eckhardt, 2008; Gonzales et al., 2009; Helfer et al., 2024; Latuamury et al., 2022; Minea, 2017; Narimani et al., 2023; Shao et al., 2020; Xie et al., 2020). Its main disadvantage, however, is its dependence on two parameters – recession constant and $BFI_{max}$. $BFI_{max}$ values are often chosen based on hydrogeological characteristics of catchments, with empirical determinations suggesting values between 0.70-0.80 for perennial streams with porous aquifers, 0.50 for ephemeral streams, and 0.20-0.25 for hard rock aquifers (Eckhardt, 2005, 2008). Baseflow estimation is often influenced by this arbitrary selection of $BFI_{max}$, leading to uncertainties and inaccuracies, particularly because the filter is highly sensitive to this parameter (Eckhardt, 2012; Narimani et al., 2023; Okello et al., 2018; Zhang et al., 2017). Studies highlight the importance of calibrating $BFI_{max}$ against other flow separation methods or tracer-based data, and have demonstrated that site-specific calibration can improve the algorithm's performance significantly (Gonzales et al., 2009; Helfer et al., 2024; Kouanda et al., 2018; Su et al., 2016; Zhang et al., 2013). This need for calibration is further supported by the fact that $BFI$ is influenced by various factors beyond hydrogeology. These include, for example, precipitation and evapotranspiration, and relationships between the two, such as the aridity index (Beck et al., 2013; Haberlandt et al., 2001; Lacey & Grayson, 1998; Mwakalila et al., 2002; Wu et al., 2019; Yao et al., 2021), soil properties such as infiltration, drainage capacity, water storage capacity and hydraulic conductivity (Ahiablame et al., 2013; Beck et al., 2013; Bloomfield et al., 2009; Longobardi & Villani, 2023; Yao et al., 2021), topographic factors such as slope, area, drainage density, altitude (Beck et al., 2013; Haberlandt et al., 2001; Lacey & Grayson, 1998; Mazvimavi et al., 2004; Mehaiguene et al., 2012; Mwakalila et al., 2002), land use and occupation (Lacey & Grayson, 1998; Mazvimavi et al., 2004), and seasonality (Longobardi & Villani,

2023; Sun et al., 2023). Therefore, it is essential to calibrate $BFI_{max}$ against other methods and data sources to ensure accurate and reliable baseflow estimation.

The calibration of $BFI_{max}$ can be effectively achieved by leveraging baseflow values derived from tracers and the mass balance equation. Among tracers, dissolved silica (DSi) has emerged as a particularly effective tool for determining baseflow values, as demonstrated in studies such as Helfer et al. (2024) and Gonzales et al. (2009). DSi, a stable and easily detectable product of rock weathering (Asano et al., 2003; Cook, 2015; Kennedy, 1971; Laudon & Slaymaker, 1997; Stewart et al., 2007), offers distinct advantages due to its natural abundance and its ability to differentiate between flow components. Pre-event groundwater ("old water"), influenced by prolonged water-rock interactions, typically exhibits high DSi concentrations (Hendershot et al., 1992; Marçais et al., 2018; Scanlon et al., 2001), while surface runoff and soil interflow (event waters, or "new water") are characterized by much lower levels (Beighley et al., 2005; Hendershot et al., 1992; Scanlon et al., 2001). These contrasting DSi concentrations make it an ideal tracer for distinguishing baseflow from quickflow, thereby enabling the refinement of RDF parameters. During the calibration of $BFI_{max}$, an objective function can be applied to minimize discrepancies between filter-derived and tracer-derived baseflow values, optimizing the $BFI_{max}$ parameter for improved accuracy (e.g., Helfer et al., 2024; Gonzales et al., 2009).

The Collischonn and Fan (2013) method offers an alternative approach for determining $BFI_{max}$ in scenarios where tracer data is unavailable. This technique relies exclusively on streamflow data and the recession constant and can be easily applied and automated. However, its effectiveness is yet to be confirmed through comparison with tracer-based studies. The method's credibility is currently grounded more in theoretical assumption than in concrete, empirical validation. Additionally, preliminary findings from Helfer et al. (2024) indicate that the method might produce inaccurate baseflow separation outcomes if the recession of interflow is not significantly faster than that of baseflow.

Taking the aforementioned factors into account, the overarching aim of this study is to improve the understanding and knowledge of the calibration and application of three baseflow separation filters. In the context of a small (< 1.2 km$^2$) rural catchment located in the south of Brazil, the specific objectives are:

1) To calibrate the $BFI_{max}$ and *Beta* parameters for Eckhardt's and Lyne and Hollick's (LH) filters. This calibration will utilize a method that aims to reduce the percentage bias between the baseflow estimates produced by these filters and those obtained from dissolved silica concentration analyses.

2) To assess and contrast the performance of the calibrated Eckhardt's and LH filters alongside the Chapman and Maxwell (CM) filter in generating baseflow hydrographs for the catchment. The evaluation will focus on the hydrographs' shapes, the timing of baseflow peaks, and baseflow to streamflow ratio.

3) To examine the performance of the Eckhardt's and LH filters using an 'event calibration approach'. This method involves determining three distinct $BFI_{max}$ and *Beta* values, each associated with the intensity range of rainfall-runoff events recorded in the study area.

## 2. Methodology

### 2.1. Study site and data collection

The Arvorezinha catchment is located in Brazil's southern region (**Fig. 1**), at coordinates 28°49'35"S 52°12'30"W. Its drainage area is 1.2 km² and its primary creek – the Lajeado Ferreira – is a perennial stream with an average channel slope of 9%. It is made of two perennial branches, one with a length of 1.5 km, and the other with 1.6 km. The creek joins the Guapore River, which is a tributary of the important Taquari River, within the South Atlantic Hydrographic basin.

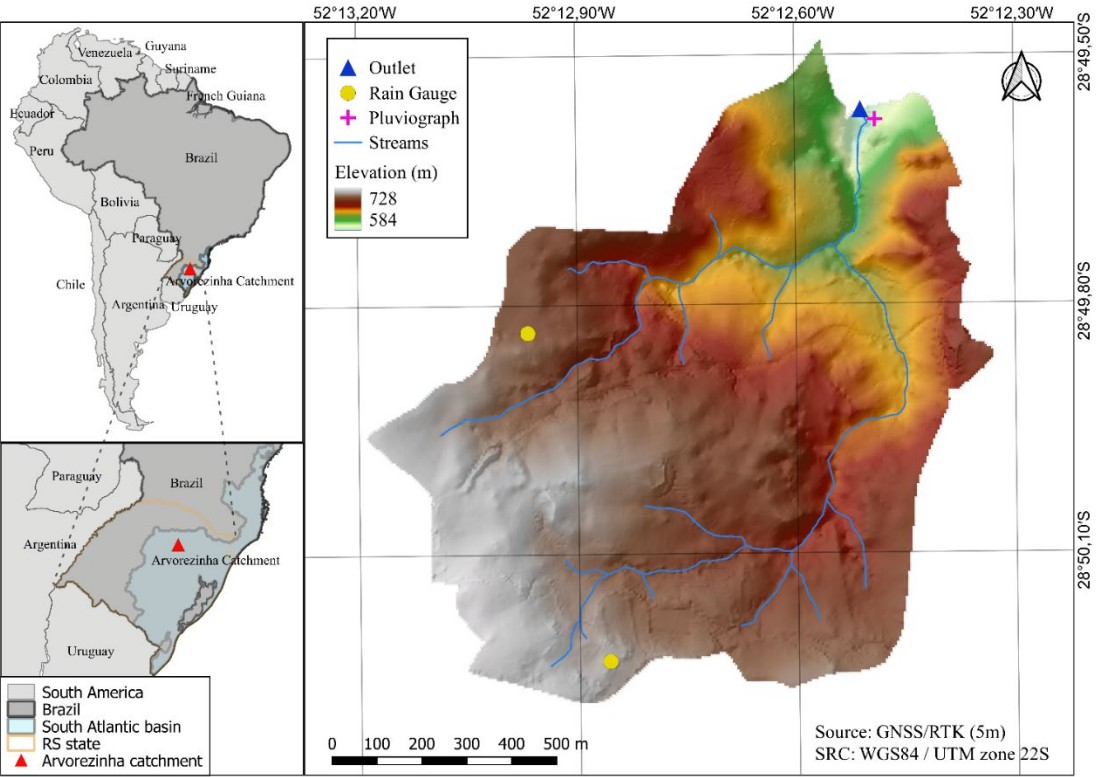

**Figure 1:** The Arvorezinha catchment and its location in southern Brazil, along with its monitoring network and elevation map. Rain gauges are indicated by yellow circles, the pluviograph, by a pink cross, and the catchment outlet, by a blue triangle.

In terms of geology, the catchment is largely made up of igneous rocks, particularly basalts and rhyodacite, and sits at elevations between 580 and 730 meters above sea level (Tiecher et al., 2017). The terrain in the upper third is relatively gentle, 155    with average slope gradients around 7%, while the middle and lower thirds are more undulated, featuring slopes with higher gradients, sometimes exceeding 15% (de Barros et al., 2014; Minella et al., 2022). Event hydrographs at the catchment outlet typically show lag times varying between 1 to 3 hours (Helfer et al., 2024).

The climate in this area is classified as Cf$_b$ (subtropical super-humid with no distinct dry season) by Köppen. Annual precipitation varies between 1250 and 2000 mm. The average annual rainfall from 2002 to 2016 was recorded at 1938 mm by Ramon (2017). Long-term monthly rainfall totals typically range from 115 mm to 170 mm, with the highest rainfall volumes and runoff intensities occurring during the spring months (September to November).

The soils in the catchment are varied: Acrisols (i.e. 'Argissolos' as per the Brazilian SiBCS Taxonomy System - (Santos et al., 2018)) make up 60% and are primarily found in the upper parts, Leptosols (i.e., 'Neossolos Litólicos') constitute 33% and are located in the lower areas, while Cambisols ('Cambissolos') account for 7% and are interspersed among the other soil types (de Barros et al., 2021; Silva et al., 2021). Leptosols and Cambisols are shallow, with a sandy top layer and no subsurface horizon, lying directly above the igneous bedrock. This structure allows for high infiltration but limited water retention, leading to quick surface and sub-surface runoff, except in areas with fractured bedrock. Acrisols, on the other hand, are deep soils and have a sharp texture change between the A and B horizons, with the B horizon containing more clay and lower infiltration capacity, resulting in slower infiltration and higher moisture retention in the top layer.

The catchment is predominantly rural. Farm sizes range from 5 to 20 hectares, often with low-tech farming practices and diverse land use (de Barros et al., 2021). The main land uses include tobacco farming (13.1%), soybean and maize crops (24%), eucalyptus plantations (34.8%), pastures and grasslands (5.2%), native forests (15.5%), and other varied uses (7.4%) (Ramon, 2017).

The Arvorezinha catchment has been a center for hydrological and erosive process research since 2002, equipped with an extensive hydrological monitoring network to study sediment generation, contamination and soil conservation practices (e.g. de Barros et al., 2021; de Barros et al., 2014; Merten & Minella, 2005; Minella et al., 2022; Ramon et al., 2017; Silva et al., 2021; Tiecher et al., 2021; Uzeika et al., 2012). For precipitation measurement, the setup includes a pluviograph providing 10-minute interval data and two pluviometers recording daily totals. Streamflow data are acquired using a water level encoder and a flume with a 1.829 m throat width. Water level readings, taken every 10 minutes, are converted to flow rates using a stage-discharge rating curve specifically calibrated for this flume.

Relevant to this study, dissolved silica concentrations (DSi) were determined from water samples collected at the catchment's outlet using a USDH-48 sampler (USGS, 1965). Although the overall monitoring of DSi spanned from 2010 to 2015, encompassing over 250 rainfall events with peak flows between 5.0 l.s$^{-1}$ and 5,000 l.s$^{-1}$, intensive DSi and flow monitoring focused on just 15 specific events (listed in **Table 1**). These 15 events were chosen to capture a range of storm characteristics, including varying rainfall intensities, durations, and antecedent moisture conditions. Furthermore, the selected events had a complete and reliable streamflow and tracer datasets. Events with missing data or poor-quality measurements were excluded to maintain the integrity of the analysis. During these selected events, DSi sampling frequency varied from every 30 minutes to 4 hours based on the event characteristics, enabling the identification of crucial baseflow hydrograph points such as the pre-

event level, peak, and recession. Detailed methodologies for DSi analysis are documented in the works of de Barros (2016) and de Barros et al. (2021).

**Table 1:** Characteristics of the 15 rainfall-runoff events analyzed in this study for baseflow hydrograph determination using DSi measurements.

| Event | Date | Number of DSi samples | $Q_{max}$ (m³ s⁻¹) | $Q_{min}$ (m³ s⁻¹) | $P_t$ (mm) | $M_i$ (mm.h⁻¹) | $D$ (h) |
|---|---|---|---|---|---|---|---|
| 1 | 17/07/2011 | 14 | 3277.6 | 0.054 | 107.7 | 5.82 | 18.5 |
| 2 | 01/10/2011 | 7 | 56.0 | 0.015 | 20.4 | 6.12 | 3.3 |
| 3 | 06/07/2012 | 9 | 518.4 | 0.003 | 105.0 | 4.63 | 22.7 |
| 4 | 26/08/2012 | 4 | 43.2 | 0.007 | 36.1 | 1.85 | 19.5 |
| 5 | 09/09/2012 | 6 | 35.3 | 0.012 | 27.0 | 5.22 | 5.2 |
| 6 | 12/03/2013 | 8 | 230.8 | 0.044 | 32.1 | 5.35 | 6.0 |
| 7 | 23/09/2013 | 6 | 532.4 | 0.031 | 37.1 | 7.68 | 4.8 |
| 8 | 26/10/2013 | 10 | 445.3 | 0.043 | 35.2 | 8.45 | 4.2 |
| 9 | 19/03/2014 | 3 | 125.1 | 0.018 | 34.3 | 4.79 | 7.2 |
| 10 | 23/07/2014 | 13 | 493.0 | 0.027 | 55.2 | 6.25 | 8.8 |
| 11 | 07/07/2015 | 11 | 535.1 | 0.080 | 48.7 | 2.21 | 22.0 |
| 12 | 12/07/2015 | 6 | 195.8 | 0.048 | 15.4 | 2.71 | 5.7 |
| 13 | 13/07/2015 | 15 | 582.7 | 0.079 | 57.2 | 5.20 | 11.0 |
| 14 | 23/07/2014 | 6 | 998.0 | 0.266 | 28.1 | 4.97 | 5.7 |
| 15 | 15/07/2015 | 3 | 353.2 | 0.190 | 12.1 | 1.54 | 7.8 |

Notes: $Q_{max}$ refers to the peak streamflow observed within the event, captured in a 10-min interval dataset, while $Q_{min}$ reflects the lowest streamflow recorded. $P_t$ denotes the total rainfall throughout the event, $M_i$ represents the average rainfall intensity during that period, and $D$ indicates the duration of the event, measured in hours.

The estimation of baseflow rates from DSi concentrations was carried out using a mass balance approach, represented by the following equation:

$$q \times DSi_q = qb_{DSi} \times DSi_{qb} + qs \times DSi_{qs} \tag{4}$$

where $DSi_q$ corresponds to the measured DSi concentration in the observed streamflow ($q$), $DSi_{qb}$ represents the concentration of DSi associated with ($qb_{DSi}$), and $DSi_{qs}$ is the DSi concentration in quickflow ($qs$) during rainfall events.

An important assumption in this tracer-based separation method is that quickflow ($qs$) carries an insignificant amount of DSi, meaning $DSi_{qs} \approx 0$ (de Barros, 2016; Gonzales et al., 2009). This assumption is justified by two key considerations. Firstly, since quickflow primarily originates from precipitation, its DSi content is minimal (Beighley et al., 2005; de Barros, 2016; Hugenschmidt et al., 2014; Rodhe, 1998; Wels et al., 1991). Secondly, quickflow has limited interaction with silica-enriched sediments, unlike groundwater, which undergoes prolonged contact along deeper subsurface pathways (Hendershot et al., 1992; Rodhe, 1998; Wels et al., 1991). Given this, the mass balance equation simplifies to: $qb_{DSi} = q \times \frac{DSi_q}{DSi_{qb}}$.

The DSi concentration corresponding to baseflow ($DSi_{qb}$) was determined from streamflow samples collected after extended dry periods, during which streamflow was assumed to be entirely sustained by groundwater (Beighley et al., 2005; Gonzales et al., 2009; Wels et al., 1991). This concentration exhibited stability over time (de Barros, 2016), making it a reliable indicator of the DSi signature of groundwater and, consequently, baseflow in the catchment.

The Arvorezinha catchment functions predominantly as a shallow, fast-responding hydrological system (de Barros et al., 2016; Helfer et al., 2024). Fractured basalt bedrock and varied soil depths promote both rapid infiltration and quickflow during rainfall events. However, during dry periods and low-intensity rainfall events, baseflow is primarily sustained by groundwater discharging from these fractured substrates. Previous studies (e.g. de Barros et al., 2016; Helfer et al., 2024) and our DSi data confirm that old groundwater contributes significantly to baseflow, as evidenced by stable, elevated silica concentrations during non-event flows. Quickflow contributions, carrying little to no dissolved silica, dominate during high-magnitude events, resulting in dilution effects. This conceptual understanding provides a sound basis for interpreting the baseflow separation results and evaluating RDF performance against DSi-derived baseflows.

## 2.2. Recession constant – a parameter for the Eckhardt's and CM filters

Prior to calibrating the filters using the baseflow values derived from DSi, the methodology involved determining the recession constant, $a$. The recession constant is required for both the CM and Eckhardt's filters (see Equations 2 and 3). The determination of $a$ involved establishing a linear relationship between the flow rates during recession phases at time $t$, and the flow rates at the next time step, $t+1$, following the methodology proposed by Eckhardt (2008) and Vogel and Kroll (1996), as represented by **Eq. 5**:

$$qb_{t+1} = a \times qb_t \tag{5}$$

To determine the recession constant, data from 204 hydrographs, recorded between 2010 and 2015, were considered. The onset of the recession phase was identified as occurring no sooner than two hours after the peak flow (or after the last peak for events with multiple peaks), ensuring minimal influence from quickflow. This 2-hour threshold was established through hydrograph analysis, which indicated a significant decline in quickflow within this period, consistent with the catchment's rapid response characteristics. Additionally, a sustained decline in flow for at least 1.5 hours was required to confirm the recession phase. The recession constant ($a$) was obtained by plotting $qb_{t+1}$ against $qb_t$ for all qualifying pairs, fitting a linear regression line constrained to pass through the origin ($qb_{t+1} = a \times qb_t$). For comparison and validation, a master recession curve (MRC) was derived by normalizing the recession curves of 20 randomly selected hydrographs (starting 2 hours post-peak) by dividing each streamflow value ($qb_t$) by its initial value ($qb_0$), averaging the normalized streamflow ($qb_t/qb_0$) across hourly time steps, and fitting an exponential best-fit curve ($qb_t/qb_0 = a^t$). The recession constant ($a$) from this MRC was adopted as an alternative estimate for comparison with the regression-based value.

## 2.3. Optimized $BFI_{max}$ (Eckhardt's filter) and $Beta$ (LH filter) parameters

The calibration process for the $BFI_{max}$ and $Beta$ parameters, used in the Eckhardt's and LH filters respectively, involved utilizing an objective function to determine their optimal values for the catchment across all 15 events. The goal was to identify $BFI_{max}$ and $Beta$ values that would minimize the Percent Bias ($PBias$) error. This error measured the discrepancy between the baseflow model estimates ($qb$) generated by the filters and the actual observed baseflow values ($qb_{DSi}$) derived from DSi data for the 15 events. $PBias$ represents the average bias of the model outputs, indicating whether they tend to under or overestimate the observed values. A $PBias$ of 0% indicates perfect agreement, positive values imply overestimation, and negative values indicate underestimation. $PBias$ was selected as the objective function for parameter optimization because it directly quantifies systematic bias (over- or underestimation tendencies) in baseflow volume estimates, has a clear optimal target of zero that is well-suited for root-finding algorithms, provides dimensionless percentage errors that enable comparison across events of different magnitudes, and is widely used in hydrological modeling studies for calibration purposes, ensuring consistency with established literature and performance evaluation standards.

A computational algorithm was written in MATLAB to iteratively refine the $BFI_{max}$ and $Beta$ values, searching within their respective limits ($0.001 \leq BFI_{max} \leq 0.999$ and $0.900 \leq Beta \leq 0.999$) to minimize the $PBias$ value, aiming to bring it as close to zero as possible. The $PBias$ was computed using **Eq. 6**:

$$PBias = 100 \times \frac{\sum_{i=1}^{N}(qb_i - qb_{DSi_i})}{\sum_{i=1}^{N} qb_{DSi_i}} \tag{6}$$

where $qb_i$ is the modelled baseflow computed by the filters ($qbECK$ or $qbLH$, as per Equations 1 and 3), and $qb_{DSi_i}$ is the silica-derived baseflow from the mass balance equation (**Eq. 4**).

The optimization algorithm used the bisection method. This iterative approach systematically narrows the search interval by evaluating the objective function ($PBias$) at the midpoint of successive intervals until convergence is achieved. For $BFI_{max}$ optimization, the initial search interval was set to [0.001, 0.999], while $Beta$ optimization used [0.900, 0.999], reflecting the physically meaningful parameter ranges. At each iteration, the algorithm calculated $PBias$ at the interval midpoint and replaced the appropriate endpoint based on the sign of the result: if $PBias$ was negative, the midpoint replaced the lower bound; if positive, it replaced the upper bound. This process continued until the interval width fell below the specified precision tolerance of 0.001, ensuring convergence to the global optimum where $PBias$ approaches zero. The bisection method was selected over gradient-based alternatives due to its guaranteed global convergence within bounded parameter spaces, robustness against local optima, and computational transparency in demonstrating parameter sensitivity across the full feasible range. This approach ensures reproducible, globally optimal solutions while maintaining consistency with established protocols in baseflow separation studies.

## 2.4. Optimized filters: Application and comparison

After determining the optimal $BFI_{max}$ and $Beta$ parameters, the Eckhardt's and LH filters (**Eq. 1** and **Eq. 3**), alongside the CM filter (**Eq. 2**), were employed to separate baseflow from the 15 analyzed rainfall-runoff events (**Table 1**). After the separation, various performance metrics such as the Nash-Sutcliffe Efficiency (*NSE*, **Eq. 7**), Kling-Gupta Efficiency (*KGE*, **Eq. 8**), and Normalized Root Mean Square Deviation (*NRSMD*, **Eq. 10**) were calculated using modelled and silica-derived baseflows to evaluate the effectiveness of the models.

*NSE* (Nash & Sutcliffe, 1970) and *KGE* (Gupta et al., 2009) range from -∞ to 1, with 1 indicating perfect agreement. For *NSE*, values >0 are minimally acceptable (Gupta et al., 1999), >0.5 are good (Partington et al., 2011; Su et al., 2016), and >0.8 are excellent for daily flows (Gupta et al., 1999; Moriasi et al., 2007; Moriasi et al., 2015). Negative values indicate poor performance (Knoben et al., 2019; Moriasi et al., 2007). For *KGE*, values >-0.41 suggest better performance than a constant baseflow, while values <-0.41 are inadequate (Gupta et al., 2009; Knoben et al., 2019).

$$NSE = 1 - \frac{\sum_{i=1}^{N}\left(qb_{DSi_i} - qb_i\right)^2}{\sum_{i=1}^{N}\left(qb_{DSi_i} - mean\left(qb_{DSi_i}\right)\right)} \tag{7}$$

$$KGE = 1 - \sqrt{(r-1)^2 + (\gamma-1)^2 + (\lambda-1)^2} \tag{8}$$

where $r$ is the Pearson correlation coefficient; $\gamma$ denotes the ratio of the standard deviation of the modeled data to that of the observed data; and $\lambda$ is the ratio of the mean of the modeled data to the mean of the observed data.

*NRMSD* (**Eq. 10**) values range from 0 to positive infinity, where lower values denote improved model accuracy (perfect match = 0). In this research, the *NRMSD* was calculated by dividing the *RMSD* (**Eq. 9**) by the observed baseflows' range, to standardize comparisons across events. *NRMSD* values below 20% were considered indicative of satisfactory model performance.

$$RMSD = \left(\frac{\sum_{i=1}^{N}(qb_i - qb_{DSi_i})^2}{N}\right)^{\frac{1}{2}} \tag{9}$$

$$NRMSD = \frac{RMSD}{qb_{DSi(max)} - qb_{DSi(min)}} \tag{10}$$

## 2.5. Event magnitude classification

One of the objectives of this study was the calibration of the $BFI_{max}$ and $Beta$ parameters (Eckhardt's and LH filters, respectively) as a function of event magnitude. This calibration aimed to understand how the parameter values varied in

response to events of differing intensities within the catchment. Before implementing this calibration, it was necessary to develop a method for classifying the events by their magnitude.

We determined three classes based on the average recurrence intervals (ARI) of the precipitation events monitored in this study (**Table 1**). The classification strategy consisted of separating the hydrographs from precipitation events with ARI less than 0.33 years, between 0.33 and 2 years, and above 2 years. This classification is because ARI > 2 years is commonly used for small hydraulic works and represents extreme values. An ARI of 0.33 can be a divider of medium and small events as it represents an event magnitude that occurs up to three times a year. It is important to highlight that the monitoring period was

only 2010 to 2015, which did not allow for a statistical projection of flow and association with their ARIs. Therefore, we established this classification subjectively.

    Therefore, the maximum flows ($Q_{max}$) produced in the 254 precipitation and flow events during the monitoring period 2010 to 2015 were grouped for the respective precipitation classes with ARI < 0.33 year (low magnitude), $0.33 \leq$ ARI $\leq 2$ years (medium magnitude), ARI $\geq 2$ years (high magnitude). The ARI of the 15 rainfall events was determined using the empirical

equation developed by Bell (1969) with parameters determined for the southern region of Brazil by Basso et al. (2019). The precipitation with a duration of one day and ARI = 10 years (a parameter required for the ARI equation) was 133.48 mm obtained through the Gumbel distribution of historical (1951-2012) rainfall series provided by the Brazilian National Water Agency, gauge station code 2852014, located at latitude -28.93 and longitude -52.13, about 12 km from the study area.

### 2.6. Optimized $BFI_{max}$ (Eckhardt's filter) and $Beta$ (LH filter) parameters per event magnitude

Following the categorization of the 15 monitored events into the three above-mentioned event magnitude classes, the calibration of the $BFI_{max}$ and $Beta$ parameters was conducted using the methodology outlined in the general calibration process (**Section 2.3**). However, the optimization algorithm was executed separately for each of the three event categories, resulting in three distinct calibrations for each filter. The objective was to achieve three sets of $BFI_{max}$ and $Beta$ values, one for each event class, that would reduce the $PBias$ error between the baseflow calculated by the filters and the actual observed baseflow

obtained from DSi data.

### 2.7. Optimized filters with event-based calibration: Application and comparison

    After the optimization of the $BFI_{max}$ and $Beta$ parameters for each event magnitude class, the Eckhardt's and LH filters, adjusted with these new parameters, were applied to determine the baseflow hydrographs for the 15 rainfall-runoff events. After this separation, evaluation metrics, including $NSE$, $KGE$, and $NRMSD$, were calculated. These metrics facilitated a comparative

analysis between the general calibration and the event-specific calibration processes, as well as a comparison among the three studied filters. This comparative analysis included examining the hydrograph shape, the timing of the baseflow peak, and the baseflow to total streamflow ratio, alongside a detailed review of similar values reported in the literature for catchments with comparable characteristics.

## 3. Results and Discussion

### 325   3.1. Recession constant – a parameter for the Eckhardt's and CM filters

The linear regression analysis (Section 2.2) resulted in a recession constant of 0.952 ( $\approx 0.0492$ h$^{-1}$). This recession constant was estimated using 204 observed hydrographs, yielding over 17,000 pairs of $qb_{t+1}$ and $qb_t$. A linear regression of $qb_{t+1}$ versus $qb_t$, forced through the origin, yielded a slope of 0.952 ($\approx 0.0492$ h$^{-1}$). A very close value (0.949) was found through an MRC analysis using 20 events of varying magnitudes, which revealed recession constants ranging from approximately 0.92 to 0.97.

The consistency of the exponential decay pattern across events supported the appropriateness of using a single representative recession constant for our catchment-scale analysis. Figure S1 in the Supplementary Material section shows the MRC results.Baseflow recession rates typically range from 0.930 to 0.995 (Jain, 2011), positioning a 0.952 value as indicative of rapid baseflow recession, corresponding to a decay rate of approximately 0.05 h$^{-1}$. The steep catchment slopes, high soil permeability, and basaltic geology explain this rapid recession in the Arvorezinha catchment. Additionally, the value of 0.952

is in line with Beck et al. (2013) and Santarosa et al. (2023), who reported similar, rapid recession constants across different Brazilian climates and catchment sizes, confirming the influence of regional geophysical characteristics on baseflow recession rates. The recession constant 0.952, found for the Arvorezinha catchment, was deemed adequate, and it was then used as an input parameter in the Eckhardt's and CM filters.

### 3.2. Optimized $BFI_{max}$ (Eckhardt's filter) and $Beta$ (LH filter) parameters

The optimization process aiming at minimizing the $PBias$ between modeled and observed baseflow values (derived from DSi) for 15 events, resulted in an optimal $BFI_{max}$ value of 0.653 for the Eckhardt's filter and 0.965 for the $Beta$ parameter of the LH filter. For both models, the optimization effectively achieved a $PBias$ close to 0.

The determined $BFI_{max}$ value of 0.653 is in close agreement with findings from studies on small catchments, such as Zhang et al. (2013), who found a $BFI_{max}$ of 0.62 for a 14.5 km² catchment with shallow soils, and Stewart (2015), who reported a $BFI_{max}$

of 0.69 for a 2.18 km² catchment with varied soil depths (0.5 m to 3 m). This value also aligns with the results by Eckhardt (2008) across 65 U.S. catchments, and by Okello et al. (2018) for catchments in South Africa of medium size. A $BFI_{max}$ value of 0.653 indicates that the model assumes the maximum long-term baseflow contribution to total flow to be 65.3%. It is important to note, however, that this value is significantly larger than the arbitrary value of 0.25 suggested by Eckhardt (2005) for catchments with hydrogeological characteristics similar to the one used in our study.

The optimum $Beta$ parameter of 0.965 diverges from the commonly referenced value of 0.925 reported in the literature (Arnold et al., 1995; Nathan & McMahon, 1990). This conventional value is often adopted globally in the absence of specific catchment data; however, there is an indication that a larger value usually yields better separation (CSIRO & SKM, 2010; Ladson et al., 2013; Li et al., 2013; Zhang et al., 2017), which is in alignment with the finding of our study.

### 3.3. Optimized filters: Application and comparison

**Figure 2** shows the results of the three separation methods investigated in this study. As described in the previous section, the formulas of Eckhardt (**Eq. 3**) and LH (**Eq. 1**) were calibrated for the parameters $BFI_{max}$ and $Beta$, respectively, using the $PBias$ minimization approach, before their application in the baseflow separation of the 15 events examined in this study (**Table 1**). Conversely, the CM method was implemented directly via **Eq. 2**, devoid of any calibration requirement for fitting parameters. This method relies exclusively on the recession constant, a parameter considered constant for the catchment.

Visually inspecting **Fig. 2**, it can be seen that the Eckhardt's (dark blue) and the CM (cyan) models yielded plausible baseflow hydrographs, both in terms of their shape and the timing of their peaks. These models accurately follow the overall smoothness observed in the streamflow hydrographs and exhibit a logical delayed peak when compared to the streamflow hydrograph's peak. Specifically, the baseflow peak occurs, on average, 1.5 hours later for the Eckhardt's filter and 1.8 hours later for the CM filter, in relation to the streamflow peak (noting that the second peak was considered for events featuring dual peaks). In 365 contrast, the LH model's baseflows increase more gradually, reaching their peak near the end of the direct runoff hydrographs and thus demonstrating a significantly longer delay in peak timing than the other two methods, with an average delay of 4.4 hours across the 15 events. It is also noteworthy that the baseflow hydrographs from the LH method converge with the streamflow hydrographs much before the rainfall-runoff event ends, a behavior that appears unrealistic.

**Table 2** shows the performance indicators $PBias$, $NSE$, $KGE$, and $NRMSD$ computed for the three methods. These metrics 370 were calculated by comparing modelled baseflows with baseflow values derived from DSi data (black dots in **Fig. 2**). For the Eckhardt's filter, the performance metrics were: $PBias \approx 0$, $NSE = 0.85$, $KGE = 0.76$, and $NRMSD = 7.5\%$, all of which fall within acceptable ranges for hydrological modeling. For the LH method, the metrics were: $PBias \approx 0$, $NSE = 0.79$, $KGE = 0.74$, and $NRMSD = 8.9\%$, indicating satisfactory performance as well. However, the CM model presented $PBias = -28\%$, $NSE = 0.86$, $KGE = 0.70$, and $NRMSD = 7.2\%$, indicating a significant underestimation of baseflows highlighted by the substantial 375 negative $PBias$, exceeding the acceptable limit of +/- 25% set in this study. This result highlights a limitation of the CM method – its inability to be calibrated due to the lack of adjustable parameters, relying solely on the recession constant, which is typically considered an invariant characteristic of the catchment. Yet, some scholars suggest that this constant may vary, proposing a potential for calibration and improvement of the CM model's baseflow separation. Exploring this possibility, particularly given the CM method's accurate representation of baseflow shape and timing, warrants further research. This 380 could offer a path to enhance the CM model's performance, addressing its current shortfall in accurately estimating baseflow.

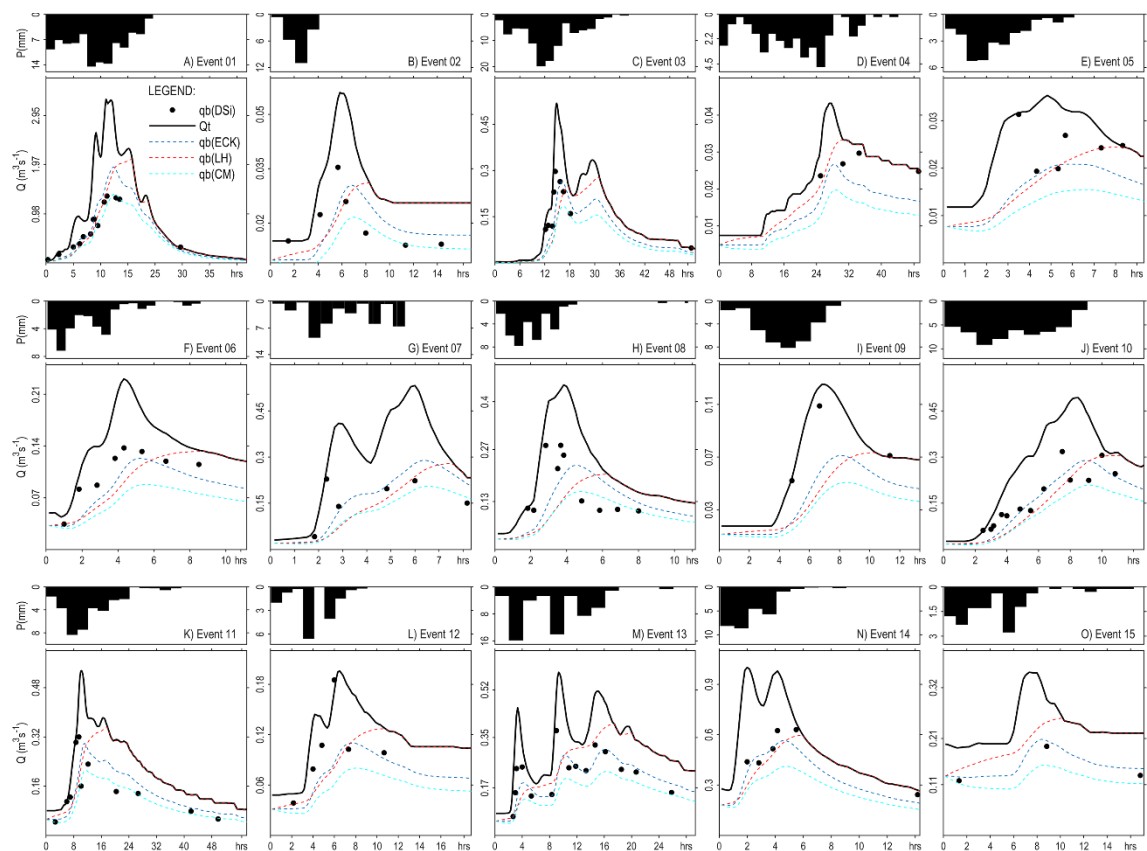

**Figure 2.** Comparison of baseflow separation techniques on 15 events in the Arvorezinha catchment (Brazil): total streamflow (*Qt*) as solid black curves; silica-derived baseflow (*qb(DSi)*) as black dots; Eckhardt's filter baseflow (*qb(ECK)*) in blue dotted lines; LH filter baseflow (*qb(LH)*) in red dotted lines; and CM filter baseflow (*qb(CM)*) in cyan dotted lines.

**Table 2.** Error metrics for modeled *versus* DSi-derived baseflows across 15 events in combination for Eckhardt's ($a = 0.952$, $BFI_{max} = 0.653$), CM ($a = 0.952$), and LH ($\beta = 0.965$) filters. Acceptable performance criteria: *NSE* > 0.5, *KGE* > -0.41, *NRMSD* < 20% and *PBias* < +/- 25%

| Filter | *PBias* (%) | *NSE* | *KGE* | *NRMSD* (%) |
|---|---|---|---|---|
| Eckhardt | < 0.05 | 0.85 | 0.76 | 7.5 |
| CM | -28 | 0.86 | 0.70 | 7.2 |
| LH | < 0.05 | 0.79 | 0.74 | 8.9 |

In terms of volume, the Eckhardt's filter showed higher volumes than the CM model. The LH model resulted in overall higher baseflow volumes than the other two models. For model comparison of volumes, we assessed the baseflow to streamflow ratios (*BF* ratio = Total Baseflow Volume / Total Streamflow Volume) for each model over 15 rainfall-runoff events, with the findings presented in **Table 3**.

**Table 3.** Comparative analysis of average *BF* ratios for the Arvorezinha Catchment (1.2 km²) in the South of Brazil across 15 rainfall-runoff events using various baseflow separation methods, alongside literature reference values.

| Separation Method | *BF* ratio (15 rainfall-runoff events) | Reference values of *BF* ratios for catchments similar to the study area or catchments located in the south of Brazil |
|---|---|---|
| Mass balance (DSi-derived baseflows) | 66%<br>CV = 0.19 | • 60-72% for rhyodacite catchments in Australia (Lacey & Grayson, 1998)<br>• 67-91% for basalt catchments in Australia (Lacey & Grayson, 1998)<br>• 50-70% for basalt catchments in the same hydrographic region in Brazil (Chagas et al., 2020)<br>• 54-65% for catchments overlying igneous aquifers in southeastern Brazil (Santarosa et al., 2023)<br>• 60-64% for small rural catchments overlying igneous aquifers in Kenya (Mugo & Sharma, 1999)<br>• 61% average *BF* ratio for six small rural catchments over granite and gneiss in Southeastern Brazil (Costa & Bacellar, 2009)<br>• 63% for catchments in Cf climate regions (Beck et al., 2013)<br>• >60% for catchments in the south of Brazil (Tan et al., 2020)<br>• >65% for catchments underlain with basalt and granite in Tanzania (Mwakalila et al., 2002)<br>• 50-70% is the predominant *BF* ratio in an analysis of 1815 undisturbed catchments in the USA. It occurs in 50% of the catchments (Xie et al., 2020) |
| Eckhardt's Filter (calibrated) ($a$ = 0.952; $BFI_{max}$ = 0.653) | 64%<br>CV = 0.04 | |
| CM Filter ($a$ = 0.952) | 49%<br>CV = 0.06 | |
| LH Filter (calibrated) ($\beta$ = 0.965) | 75%<br>CV = 0.15 | |

The *BF* ratios were determined to be 64%, 49%, and 75% for the Eckhardt's, CM, and LH filters, respectively. The *BF* ratio based on baseflows calculated from DSi was 66%, positioning the Eckhardt's filter as the most aligned with observed data. Moreover, the Eckhardt's filter exhibited the most consistent performance across the 15 events, demonstrating the lowest variability with a coefficient of variation (CV) at 0.04, as shown in **Table 3**. This consistency is crucial for accurately reflecting catchment conditions, as highlighted by Nathan and Mcmahon (1990). Furthermore, the results obtained from the Eckhardt's filter are closely aligned with several findings reported in the literature. Lacey and Grayson (1998) identified a notable trend of higher *BF* ratios in catchments underlain by igneous rocks, such as basalt and rhyodacite, compared to those underlain by sedimentary and metamorphic rocks. Their study, which included 23 catchments underlain by basalt and/or rhyodacite (reflecting the geological composition of the study catchment), reported *BF* ratios of 66% for rhyodacite and 79% for basalt, compared to an average of 54% across 114 catchments with diverse geological formation. They attributed the high *BF* ratios in igneous rock catchments to high primary porosity and the presence of fractures, which enhances groundwater recharge, storage capacity, and the formation of deep, permeable soils. A similar trend was reported by Mwakalila et al. (2002) in which larger *BFI* values were found for catchments underlain with granites and basalt in an examination of 12 catchments ranging from 66 to 2930 km² in semi-arid environments of Tanzania. Such characteristics are consistent with those of the Arvorezinha catchment, supporting the observed high baseflow proportion. Therefore, the *BF* ratios from DSi-based baseflows, along with those obtained from Eckhardt's and LH filters in the Arvorezinha catchment, were in realistic agreement with expectations,

 while the CM filter's derived *BF* ratio appeared to be an outlier. Table 3 further includes comparative *BF* values from the literature for catchments analogous to the one under study, providing a broader context for our findings.

Therefore, considering shape, peak timing, magnitude and *BF* ratios, the Eckhardt's filter emerges as the most accurate for baseflow separation in the Arvorezinha catchment. The superior performance of the Eckhardt's filter over the LH and the CM models matches the results reported by Xie et al. (2020), who evaluated nine separation methods in 1815 catchments in the USA, demonstrating higher *NSE* and *KGE* values for the Eckhardt's filter. Similarly, Gonzales et al. (2009) reached a comparable conclusion when they evaluated seven non-tracer-based separation methods against baseflow estimations derived from a tracer-based approach in a lowland catchment in the Netherlands, further reinforcing the effectiveness of the Eckhardt's filter.

**Figure 3** illustrates the relationship between modeled and observed baseflow values for the three filters. For the Eckhardt's and LH filters, it can be seen that low flows are underestimated, and high flows are overestimated. A more favorable agreement is noted for mid-range flows. These results were expected since this is a direct consequence of adjusting the $BFI_{max}$ and *Beta* parameters to reduce the average discrepancy between observed and modeled values. As such, this process leads to the models balancing underestimated values through overestimating values in other instances, and vice versa. For the CM model, the flows were generally underestimated.

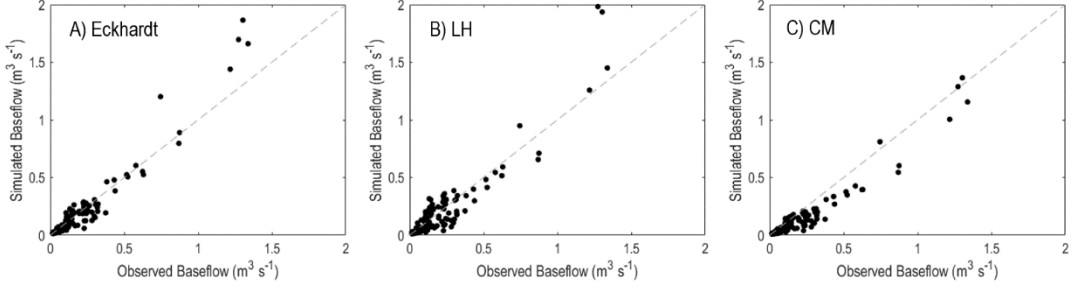

**Figure 3:** Scatter plot comparing modeled baseflow values against observed values derived from DSi, with the grey dashed line representing the 1:1 line. For the Eckhardt (A) and LH (B) models, the plots show the calibrated models' trend to overestimate higher baseflows and underestimate lower baseflows, a predictable result from the optimization method employed during calibration. The CM model (C) consistently underestimated the baseflow values.

Despite the quality indicators indicating a satisfactory match between the modeled and observed baseflows (**Table 2**), pinpointing a *Beta* value of 0.952 and a $BFI_{max}$ value of 0.653 as the optimal parameters for this specific catchment, discrepancies become evident upon examining data from separate events (refer to **Tables 4** and **5**), suggesting that improvements could be achieved with event-specific calibration. **Tables 4** and **5** show that most events did not meet all the quality indicators criteria simultaneously. This issue reveals the calibrated models' challenge in precisely capturing baseflow

in some cases, implying that a calibration process designed for each event or event of similar sizes might offer more accurate modeling than a generalized approach.

**Table 4.** Error metrics for observed *versus* modeled baseflows across 15 events, utilizing the calibrated **Eckhardt's filter** ($a$ = 0.952, $BFI_{max}$ = 0.653). Acceptable model performance criteria: $NSE > 0.50$, $KGE > -0.41$, $NRMSD < 20\%$ and $PBias < +/-$ 25%

| Event | $Q_{max}$ (l.s$^{-1}$) | DSi samples | PBias (%) | NSE | KGE | NRMSD (%) | BF ratio |
|---|---|---|---|---|---|---|---|
| 1 | 3277.6 | 14 | 17.7 | 0.66 | 0.55 | 19.6 | 0.65 |
| 2 | 56.0 | 7 | -3.8 | 0.10 | 0.51 | 32.4 | 0.64 |
| 3 | 518.4 | 9 | -21.3 | 0.31 | 0.69 | 26.3 | 0.65 |
| 4 | 43.2 | 4 | -24.6 | -4.50 | 0.52 | 111.1 | 0.64 |
| 5 | 35.3 | 6 | -22.3 | -2.41 | -0.65 | 63.0 | 0.61 |
| 6 | 230.8 | 8 | -19.5 | 0.45 | 0.79 | 23.2 | 0.61 |
| 7 | 532.4 | 6 | -7.5 | -0.49 | 0.38 | 41.7 | 0.56 |
| 8 | 445.3 | 10 | -13.3 | -0.23 | 0.26 | 46.7 | 0.62 |
| 9 | 125.1 | 3 | -43.9 | -1.37 | 0.44 | 64.0 | 0.61 |
| 10 | 493.0 | 13 | -9.6 | 0.77 | 0.83 | 15.9 | 0.59 |
| 11 | 535.1 | 11 | -4.2 | 0.33 | 0.61 | 26.3 | 0.65 |
| 12 | 195.8 | 6 | -27.7 | -0.04 | 0.37 | 30.5 | 0.63 |
| 13 | 582.7 | 15 | -10.1 | 0.33 | 0.65 | 20.4 | 0.64 |
| 14 | 998.0 | 6 | -8.5 | 0.76 | 0.87 | 16.7 | 0.65 |
| 15 | 353.2 | 3 | 8.03 | 0.86 | 0.92 | 16.3 | 0.65 |
| *All events combined* | 121 | | < 0.05 | 0.85 | 0.74 | 7.5 | 0.64 |

**Table 5.** Error metrics for observed *versus* modeled baseflows across 15 events, utilizing the calibrated **LH filter** ($\beta$ = 0.965). Acceptable model performance criteria: $NSE > 0.50$, $KGE > -0.41$, $NRMSD < 20\%$ and $PBias < +/-$ 25%

| Event | $Q_{max}$ (l.s$^{-1}$) | DSi samples | PBias (%) | NSE | KGE | NRMSD (%) | BF ratio |
|---|---|---|---|---|---|---|---|
| 1 | 3277.6 | 14 | 8.7 | 0.58 | 0.52 | 21.7 | 0.71 |
| 2 | 56.0 | 7 | 8.3 | -0.89 | -0.03 | 47.1 | 0.79 |
| 3 | 518.4 | 9 | -34.8 | -0.48 | 0.43 | 38.5 | 0.78 |
| 4 | 43.2 | 4 | 6.3 | -1.47 | -0.17 | 59.6 | 0.88 |
| 5 | 35.3 | 6 | -16.8 | -2.28 | -0.28 | 61.9 | 0.67 |
| 6 | 230.8 | 8 | -18.9 | 0.08 | 0.67 | 30.0 | 0.69 |
| 7 | 532.4 | 6 | -21.5 | -1.00 | 0.29 | 48.4 | 0.50 |
| 8 | 445.3 | 10 | -25.4 | -1.17 | -0.24 | 61.9 | 0.63 |
| 9 | 125.1 | 3 | -42.1 | -2.10 | 0.20 | 73.2 | 0.67 |
| 10 | 493.0 | 13 | -15.4 | 0.61 | 0.69 | 20.9 | 0.60 |
| 11 | 535.1 | 11 | 9.2 | -0.18 | 0.38 | 35.0 | 0.88 |
| 12 | 195.8 | 6 | -23.7 | -0.25 | 0.29 | 33.4 | 0.79 |
| 13 | 582.7 | 15 | 7.7 | -0.49 | 0.39 | 30.3 | 0.82 |
| 14 | 998.0 | 6 | -11.3 | 0.66 | 0.81 | 19.9 | 0.74 |
| 15 | 353.2 | 3 | 38.2 | -2.66 | 0.44 | 84.1 | 0.85 |
| *All events combined* | 121 | | < 0.05 | 0.79 | 0.76 | 8.9 | 0.75 |

### 3.4. Event magnitude classification

The violin diagrams in **Fig. 4** display the distribution of the peak flow rates from 254 rainfall-runoff events observed during the monitoring period (2010-2015) into the classes 'low' (precipitation with ARI < 0.33 year), 'medium' (precipitation with 0.33 ≤ ARI ≤ 2 years), and 'high' (precipitation with ARI > 2 years). The stream flow rate thresholds to distinguish the three classes were selected based on the interquartile ranges of the three classes. As such, hydrographs with peak flow rates < 250 $\ell$ s$^{-1}$ (= third quartile of the first class) were classified as 'low' magnitude. Hydrographs with peak flow rates between 250 $\ell$ s$^{-1}$ and 1000 $\ell$ s$^{-1}$ (= third quartile of the second class) were classified as 'medium' magnitude. Hydrographs with peak flow rates higher than 1000 $\ell$ s$^{-1}$ were classified as 'high' magnitude.

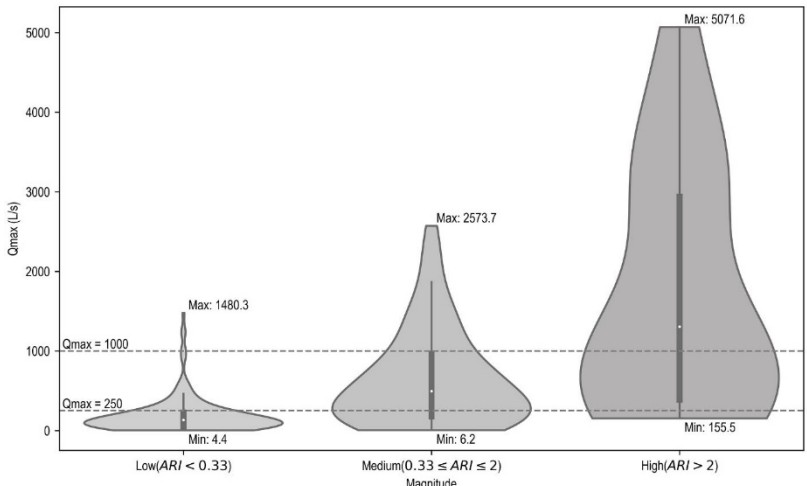

**Figure 4:** Distribution of peak flow rates from 254 rainfall-runoff events observed during the monitoring period (2010-2015) into the classes 'low', 'medium', and 'high' magnitudes.

The results of the distribution of events within each class of event magnitude, excluding the outliers, is presented in **Table 6**. The number of events observed in each class over the six-year data series period (2010 to 2015) is also shown, highlighting a higher occurrence of low-magnitude events (134) compared to medium (61) and high magnitudes (33).

**Table 6:** Distribution of the 15 monitored rainfall-runoff into the classes 'low', 'medium', and 'high' magnitudes.

| Class | Peak magnitude | Nº of events between 2010 and 2015[1] | DSi events[2] | DSi samples |
|---|---|---|---|---|
| Low | $< 250 \, \ell \, s^{-1}$ | 134 | 2,4,5,6,9,12 | 34 |
| Medium | $250 - 1000 \, \ell \, s^{-1}$ | 61 | 3,7,8,10,11,13,14,15 | 73 |
| High | $> 1000 \, \ell \, s^{-1}$ | 33 | 1 | 14 |

[1] Denotes the total number of events observed in the period 2010-2015. [2] Denotes the events used for analysis due to the availability of DSi data, as listed in **Table 1**.

### 3.5. Optimized $BFI_{max}$ (Eckhardt's filter) and *Beta* (LH filter) parameters per event magnitude

The optimization process, aimed at reducing the *PBias* between modeled and observed baseflow values (derived from DSi) for each event magnitude category listed in **Table 6**, yielded three significantly distinct $BFI_{max}$ values:

- 0.809 for 'low' magnitude events;
- 0.701 for 'medium' magnitude events; and
- 0.576 for 'high' magnitude events.

These values show a notable deviation from the average $BFI_{max}$ value of 0.653 established for the catchment, suggesting the $BFI_{max}$ parameter's sensitivity to the magnitude of rainfall-runoff events. The calibration for each event category achieved *PBias* values near zero. Like the general calibration, the $BFI_{max}$ values derived per event magnitude also exceeded the arbitrary value of 0.25 recommended by Eckhardt (2005) for catchments of hydrogeological characteristics like the Arvorezinha's, supporting the study's hypothesis that events of low magnitude align better with higher $BFI_{max}$ values, whereas those of high magnitude align better with lower values. This observation is consistent with findings by Minea (2017) and Okello et al. (2018), who reported higher $BFI_{max}$ values during low-flow (dry) seasons and lower values during high-flow (wet) seasons. Zhang et al. (2013) also noted a similar trend with larger baseflow indexes in dry seasons compared to wet seasons in a Canadian catchment.

The variation in optimum $BFI_{max}$ values across event magnitudes is explained by changes in the relative contribution of baseflow to total streamflow as a function of event size. During low-magnitude events, streamflow is predominantly sustained by perennial groundwater discharge, resulting in a high optimum $BFI_{max}$ value of 0.809. This indicates that baseflow constitutes the majority of flow under dry or low-intensity rainfall conditions. As event magnitude increases, the proportion of quickflow rises due to enhanced surface and near-surface runoff, which progressively dilutes the baseflow component. This shift is reflected in lower $BFI_{max}$ values for medium (0.701) and high-magnitude (0.576) events. These decreasing values align with expectations for steep, fractured catchments like Arvorezinha, where intense storms can trigger rapid hydrological responses with minimal groundwater interaction. These findings suggest that adopting tiered calibration can capture underlying hydrological processes, including event-driven shifts in groundwater–surface water connectivity.

For the *Beta* parameter of the LH filter, optimal values were identified as:

- 0.921 for 'low' magnitude;

- 0.957 for 'medium' magnitude; and
- 0.970 for 'high' magnitude events.

These values diverge from the catchment's average *Beta* value of 0.965, indicating that this parameter is also influenced by event magnitude. Interestingly, only the calibration for 'low' magnitude events aligns with the widely recognized standard value of 0.925 referenced in the literature (Arnold & Allen, 1999; Arnold et al., 1995; Lyne & Hollick, 1979; Nathan & Mcmahon, 1990). The values determined for 'medium' and 'high' magnitude events exceed this benchmark, falling outside the commonly accepted range of 0.90 to 0.95 (Nathan & McMahon, 1990). Additionally, the *Beta* value for 'medium' magnitude events (0.957) is closely aligned with the recession constant. This finding supports the approach of using the recession constant as the most suitable parameter for calibration, a strategy endorsed by numerous researchers (e.g. Eckhardt, 2005; Tan et al., 2009b; Zhang et al., 2017; Mugo & Sharma, 1999). The highest *Beta* value identified in this study (0.970) aligns with findings by SKM and CSIRO (2012), as cited by Ladson et al. (2013), where a *Beta* value of 0.98 was deemed more appropriate for Australian catchments analyzed using tracer-based baseflow measurements.

The variation in optimum *Beta* values across event magnitudes is explained by the dynamic interplay between groundwater contributions and event-driven quickflow within the catchment. During low-magnitude events, the streamflow is primarily sustained by older groundwater, resulting in a dominant baseflow signal and a lower optimum *Beta* value (0.921), which aligns with widely accepted literature standards. As event magnitude increases, surface and shallow subsurface runoff begin to contribute more significantly to streamflow. For medium events, the *Beta* value (0.957) closely matches the catchment's recession constant, suggesting a balanced mix of baseflow and event water. In high-magnitude events, the highest *Beta* value (0.970) reflects a storm response dominated by rapid quickflow and dilution of the baseflow component — consistent with a reduced influence of older groundwater. This pattern reveals that *Beta* is not a static parameter, but one that reflects the proportion of groundwater contribution in streamflow, which varies with event size.

It is important to observe that the event-specific variation in $BFI_{max}$ and *Beta* parameters observed in this study represents genuine characteristic hydrological responses of the catchment to different flow conditions, rather than model uncertainty or parameter instability. Unlike the recession constant, which reflects the intrinsic storage-discharge relationship of the catchment and remains relatively time-invariant (0.952), the $BFI_{max}$ and *Beta* parameters capture the dynamic mixing ratios between flow components under varying hydrological conditions. The observed variation in these parameters across rainfall-runoff event magnitudes reflects how the relative dominance of baseflow versus quickflow evolves under different hydrological conditions. In low-magnitude events, baseflow is the dominant component, yielding higher $BFI_{max}$ and lower *Beta* values, while in high-magnitude events, the surge of quickflow reduces the relative contribution of baseflow, driving these parameters in the opposite direction. This physical interpretation aligns with established understanding of catchment behavior, where the recession constant describes "how fast" water drains from storage, while $BFI_{max}$ and *Beta* describe "what proportion" of flow originates from different sources under varying conditions. Therefore, these findings suggest that event-based calibration captures real and predictable hydrological behavior, which is consistent with seasonal variations reported in the literature (Minea, 2017;

Okello et al., 2018). While $BFI_{max}$ and Beta have been originally designed as flexible fitting parameters, the results of this study indicate that they may also encode characteristic features of the catchment, such as aquifer response, baseflow persistence, and subsurface connectivity. This interpretation aligns with Eckhardt's (2005) original caution about $BFI_{max}$, but also supports the idea that, with appropriate reference data (such as DSi), its values can reflect real catchment properties, not merely serve a numerical role in hydrograph separation.

### 3.6. Optimized filters with event-based calibration: Application and comparison

**Figure 5** shows the results of the three separation methods analyzed in this study, in which the Eckhardt's and the LH filters underwent calibration for $BFI_{max}$ and *Beta* parameters based on event magnitude. The method CM, on the other hand, was applied directly using **Eq. 2**, since this method does not have a fitting parameter that can be calibrated.

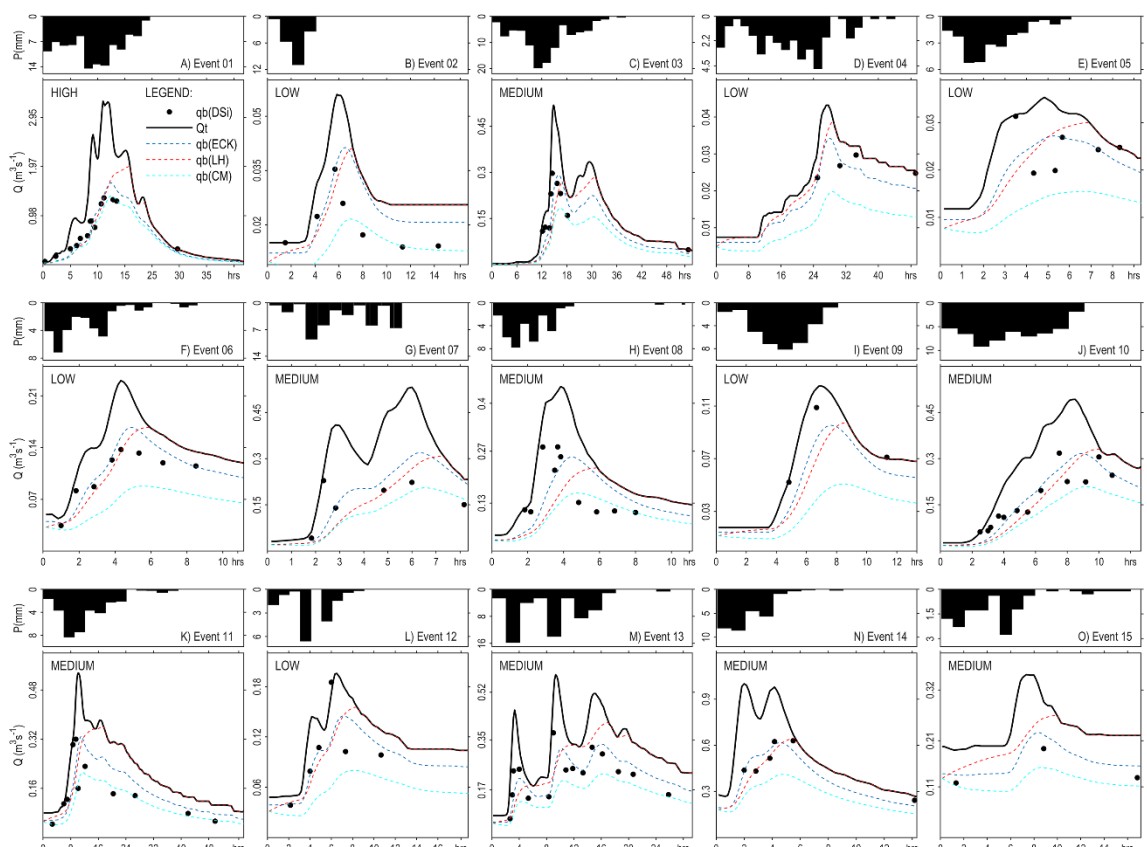

**Figure 5.** Comparison of baseflow separation techniques on 15 events in the Arvorezinha catchment (Brazil): total streamflow (*Qt*) as solid black curves; silica-derived baseflow (*qb(DSi)*) as black dots; Eckhardt's filter baseflow (*qb(ECK)*) in blue dotted lines; LH filter baseflow (*qb(LH)*) in red dotted lines; and CM filter baseflow (*qb(CM)*) in cyan dotted lines.

Note: Eckhardt's and LH filters used varying parameters as a function of event magnitude (low, medium or high).

Visual inspection of **Fig. 5** (event-based calibration results) and **Fig. 2** (general calibration results) does not allow for an accurate determination of the changes brought about by the event-based calibration, as similar observations can be noted for both calibration processes. Eckhardt's filter and CM filter yielded more plausible baseflow hydrographs in terms of shape (both follow the smoothness of the streamflow hydrographs and present a reasonably delayed peak in comparison to the peak observed on the streamflow hydrograph). In terms of volume, the Eckhardt's filter shows higher volumes than the CM model. The LH model resulted in overall higher baseflow volumes than the other two models. The baseflow estimated by the LH model increases gradually and achieves its maximum rate near the end of the direct runoff hydrograph, presenting a higher delay in baseflow peak, as compared to the other two methods. The event-based calibration led to a reduction of the delay between baseflow and streamflow peaks. For the Eckhardt's filter, the average event delay was calculated as 1.5 hours using the general calibration, which reduced to 1.3 hours with the event-based calibration. For the LH filter, the average event delay reduced from 4.4 hours (general calibration) to 3.4 hours (event-based calibration).

**Table 7** shows the performance indicators *PBias, NSE, KGE,* and *NRMSD* computed for the Eckhardt's and LH filters after the event-based calibration. These metrics were calculated comparing modelled baseflows with baseflow values derived from DSi data. The asterisks (*) show the level of improvement in the performance indicators, compared to the general calibration. The event-based calibration has significantly improved the metrics for both models.

**Table 7.** Error metrics for observed (DSi-derived) *versus* modeled baseflows considering 15 events and two separation methods adopting the following parameters: Eckhardt's filter ($a = 0.952$, $BFI_{max}^{(low)} = 0.809$, $BFI_{max}^{(medium)} = 0.701$, $BFI_{max}^{(high)} = 0.576$); LH filter ($\beta^{(low)} = 0.921$, $\beta^{(medium)} = 0.957$, $\beta^{(high)} = 0.970$). Acceptable model performance criteria: $NSE > 0.5$, $KGE > -0.41$, $NRMSD < 20\%$ and $PBias < +/- 25\%$

| Filter | *PBias* (%) | *NSE* | *KGE* | *NRMSD* (%) |
|--------|-------------|-------|-------|-------------|
| Eckhardt | < 0.50 | 0.92*** | 0.91**** | 5.4** |
| LH | < 1.20 | 0.84** | 0.85*** | 7.8* |

Improvement in relation to the general calibration: *5-20%; **20-35%; ***35-50%; ****>50%. The absence of a (*) indicates no significant improvement.

Following the event-based calibration, the *BF* ratios were computed as 65% for the Eckhardt's model and 76% for the LH model, representing a slight change from the general calibration outcomes of 64% for Eckhardt's and 75% for LH. The *BF* ratio from the Eckhardt's filter remains the most accurate, aligning more closely with the DSi-derived *BF* ratio of 66%. The *BF* ratios computed after the event-based calibration alongside reference values are summarized in **Table 8**. The observed increase in the value of CV for baseflow ratios from 0.04 (general calibration) to 0.09 (event-based calibration) for the Eckhardt's filter reflects the intentional variability introduced by assigning different $BFI_{max}$ values across rainfall-runoff event magnitudes. This higher variability in model outputs is expected and hydrologically meaningful: it reflects shifts in the dominant flow processes (from baseflow-driven in low events to quickflow-dominated in high events). In contrast, the LH filter's CV remained relatively stable (from 0.15 to 0.14) despite event-specific calibration, due to its lower sensitivity to the

*Beta* parameter. These differences suggest that while the Eckhardt's filter can dynamically respond to hydrologic variability, the LH method's structural limitations lead to a more uniform output, regardless of parameter variation.

**Table 8.** Comparative analysis of average *BF* ratios for the Arvorezinha Catchment (1.2 km²) in Southern Brazil across 15 rainfall-runoff events using various baseflow separation methods and an event-based calibration approach, alongside literature reference values.

| Separation Method | BF ratio (average of 15 rainfall-runoff events) | Reference values of BF ratios for catchments similar to the study area or catchments located in the South of Brazil |
|---|---|---|
| Mass balance (DSi-derived baseflows) | 66% CV = 0.19 | • 60-72% for rhyodacite catchments in Australia (Lacey & Grayson, 1998) |
| Eckhardt's Filter (general calibration) ($a = 0.952$; $BFI_{max} = 0.653$) | 64% CV = 0.04 | • 67-91% for basalt catchments in Australia (Lacey & Grayson, 1998) • 50-70% for basalt catchments in the same hydrographic region in Brazil (Chagas et al., 2020) |
| Eckhardt's Filter (event-based calibration) ($a = 0.952$; $BFI_{max}^{(low)} = 0.809$, $BFI_{max}^{(medium)} = 0.701$, $BFI_{max}^{(high)} = 0.576$) | 65% CV = 0.09 | • 54-65% for catchments overlying igneous aquifers in southeastern Brazil (Santarosa et al., 2023) • 60-64% for small rural catchments overlying igneous aquifers in Kenya (Mugo & Sharma, 1999) • 61% for six small rural catchments over granite and gneiss in Southeastern Brazil (Costa & Bacellar, 2009) |
| LH Filter (general calibration) ($\beta = 0.965$) | 75% CV = 0.15 | • 63% for catchments in $C_f$ climate regions (Beck et al., 2013) |
| LH Filter (event-based calibration) ($\beta^{(low)} = 0.921$, $\beta^{(medium)} = 0.957$, $\beta^{(high)} = 0.970$) | 76% CV = 0.14 | • 60% for catchments in the south of Brazil (Tan et al., 2020) • 65% for catchments underlain with basalt and granite in Tanzania (Mwakalila et al., 2002) |
| CM Filter ($a = 0.952$) | 48% CV = 0.06 | • 50-70% is the predominant *BF* ratio in an analysis of 1815 undisturbed catchments in the USA. It occurs in 50% of the catchments (Xie et al., 2020) |

**Figure 6** illustrates the relationship between modeled and observed baseflows for the Eckhardt and LH filters, comparing
results from both general and event-based calibration methods. This comparison highlights the improvement in model performance following the event-based calibration. Specifically, this approach has led to improved predictions of the data extremes, including both low and high flows, which were not accurately predicted with the general calibration process.

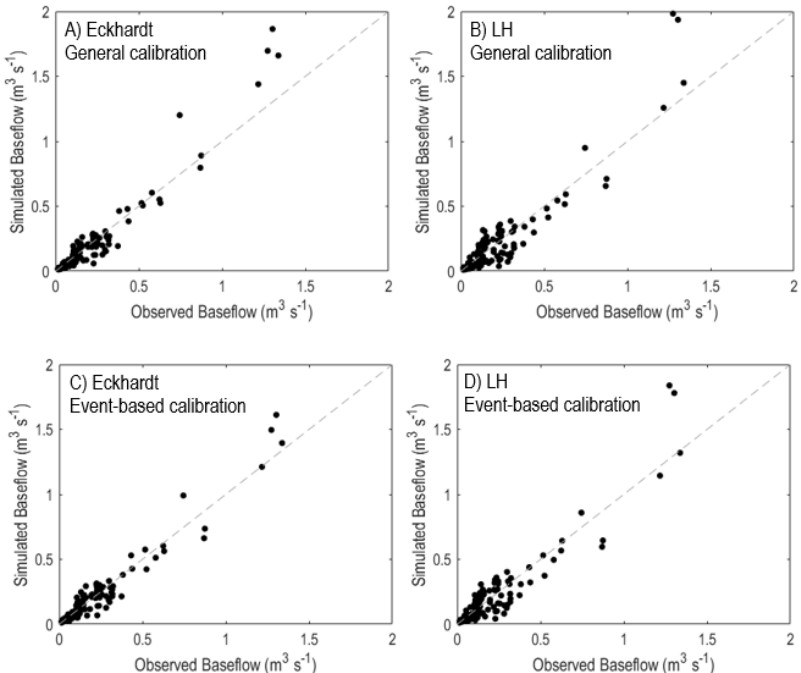

**Figure 6:** Scatter plot comparing modeled baseflow values against observed values derived from DSi, with the grey dashed line representing the 1:1 line. Eckhardt – General calibration (A), LH – General Calibration (B), Eckhardt – Event-based calibration (C), LH – Event-based Calibration (D). The plots show the higher performance of the models under event-based calibration, particularly with a better prediction oh high flows in comparison to the general calibration approach.

Analysis of individual events, as detailed in **Figures 7** and **8** and **Tables A1** and **A2** (Appendices), indicates that the event-based calibration method led to improvements in performance metrics for both the LH and Eckhardt's filters across most events. The data highlight that, for the Eckhardt's filter, the event calibration approach enhanced the *PBias* indicator significantly in 12 events, *NSE* and *NMRSD* in 10 events, and *KGE* in nine events. Regarding the LH filter, there were enhancements observed in the *PBias* for 10 events, in the *NSE* metric for 11 events, and in *NRMSD* and *KGE* in nine events. Events 5 and 9, for instance, highlight the general calibration's poor performance with extreme values in *NSE* and *NRMSD* values for both filters, which the event-based method mitigated significantly (**Figures 7** and **8**). These enhancements emphasize the advantage of tailoring calibration to specific event magnitudes, yielding more precise modeling outcomes compared to a generic calibration approach. However, some events exhibited contrasting results; Event 15, for example, displayed higher *PBias* and worse *NSE* with event-based calibration compared to the general method for both filters (**Figures 7** and **8**), indicating that event-based calibration may not always be superior. While the event-based method generally outperforms in most scenarios, event-specific characteristics can occasionally make the general calibration more suitable.

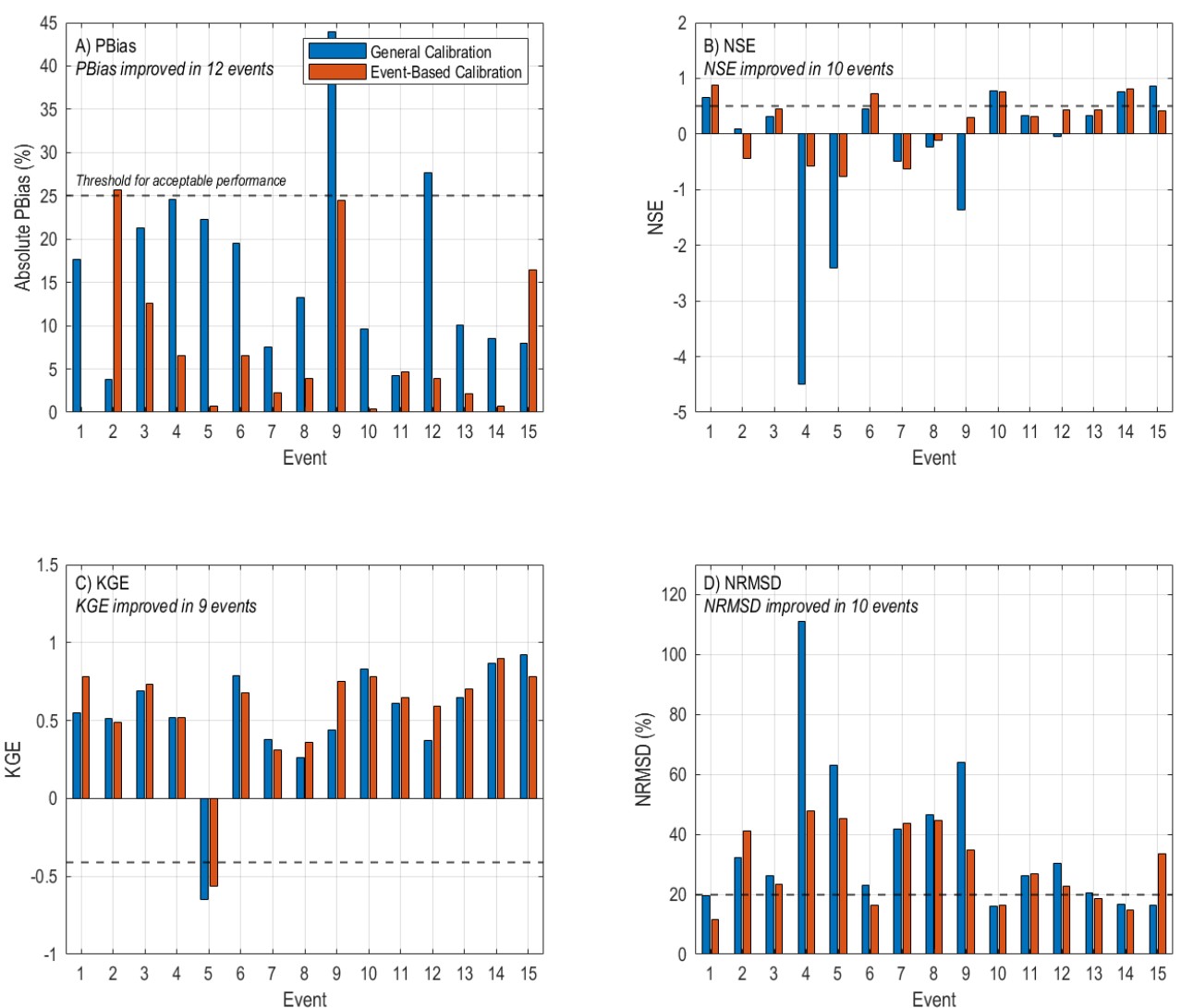

**Figure 7:** Comparison of performance metrics for general **Eckhardt's filter** calibration (blue) and event-based calibration (orange) across 15 events. (A) Absolute *PBias*, improved in 12 events; (B) *NSE*, improved in 10 events; (C) *KGE*, improved in 9 events; (D) *NRMSD*, improved in 10 events.

*595*

*600*

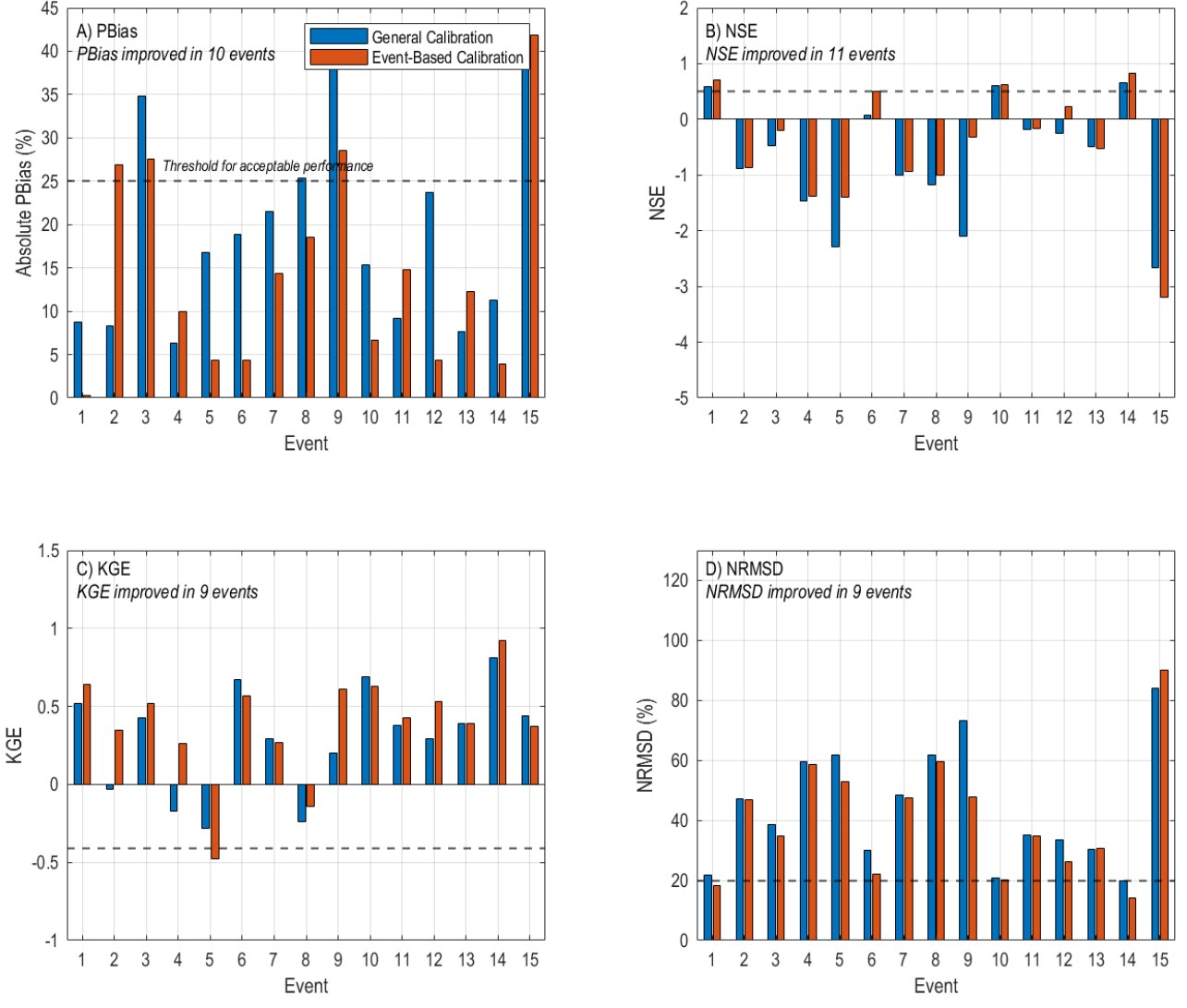

**Figure 8:** Comparison of performance metrics for general **LH filter** calibration (blue) and event-based calibration (orange) across 15 events. (A) Absolute *PBias*, improved in 10 events; (B) *NSE*, improved in 11 events; (C) *KGE*, improved in 9 events; (D) *NRMSD*, improved in 9 events.

*605*

## 4. Considerations for using the DSi mass-balance method to calibrate RDF filters

The calibration of *BFI_max* and *Beta* parameters using the DSi mass-balance approach requires careful attention due to fundamental differences in how flow components are distinguished. RDFs primarily separate baseflow and quickflow based

on flow velocity characteristics (Helfer et al., 2024), whereas the mass-balance technique relies on variations in DSi concentrations. This contrast is especially important in larger catchments, where transient water contributions, such as the mobilization of older stored water and return flow processes, can substantially impact hydrograph responses (Cartwright et al., 2018; Yang et al., 2021).

RDF filters differentiate baseflow as slow-moving components like groundwater and bank storage return flow, while identifying quickflow as rapid-response flows, including surface runoff, soil interflow, and direct precipitation. Conversely, the DSi mass-balance method classifies flow components based on DSi concentrations, designating high-DSi components (groundwater and old-water flushing) as baseflow and low-DSi components (surface runoff, return flow, and direct precipitation) as quickflow.

During storm events, these differences become particularly pronounced. Old-water flushing introduces a rapid increase in DSi concentrations early in the hydrograph, reflecting the displacement of solute-rich water stored in the soil matrix. While the mass-balance method captures this as baseflow, RDF filters, designed to represent the gradual response of slow groundwater movement, assign such flows to quickflow. Similarly, bank storage return flow, characterized by dilution effects, is considered baseflow in RDF filters but quickflow in the mass-balance method. These contrasting classifications can result in misaligned peak timings and differing baseflow dynamics, particularly in short-term hydrographs.

In the Arvorezinha catchment, where transient water effects such as old-water flushing and return flow are minimal due to its shallow, clayey soils, the DSi mass-balance method and RDF filters can be reasonably aligned for calibration. However, during some events, discrepancies were observed. For example, events 2 and 9 showed earlier baseflow peaks in the mass-balance method, likely caused by old-water flushing. This initial peak was subsequently reduced as event groundwater and return flows, with lower DSi concentrations, diluted the streamflow.

Although the mass-balance method may not be ideal for short-term baseflow separation (hourly or daily) due to transient water effects, it is still well-suited for long-term applications. Over extended periods, the calibrated filters, particularly the Eckhardt's filter, ensure cumulative baseflow estimates are consistent with those of the DSi method. To enhance the reliability of such calibrations, validating results against reported *BFI* values from the literature is strongly recommended (Helfer et al., 2024). Ultimately, the differences between the two methods highlight the need for a nuanced application of the mass-balance method
in calibrating RDF filters. Future studies could focus on combining both approaches to create multi-component streamflow separation methods or assess the impact of transient water sources, providing deeper insights into hydrological processes and flow dynamics.

## 5. Conclusions and Recommendations

This study covered a detailed examination of baseflow separation techniques within a small rural catchment, applying both general and event-based calibration strategies to optimize the parameters of Eckhardt's, Lyne and Hollick (LH), and Chapman and Maxwell (CM) filters. The initial objective of calibrating $BFI_{max}$ (Eckhardt's) and $Beta$ (LH) parameters to minimize differences between modeled and tracer-based baseflow values was successfully achieved, with both filters demonstrating improved accuracy over the CM method.

The CM model, reliant solely on streamflow and the recession constant, lacked the flexibility for fine-tuning, restricting its accuracy in volume and peak magnitude estimation, with a notable average underestimation of baseflow quantities by 28%. On the other hand, the flexibility of the adjustable parameters in the Eckhardt's and LH models enabled a more precise representation of baseflow hydrographs after calibration.

For the studied catchment – a small (1.2 km²) rural basin characterized by steep slopes, rapid baseflow recession (recession constant = 0.952), fractured basalt geology, and mixed soil types with high infiltration capacity – optimal calibration yielded $BFI_{max}$ and $Beta$ values of 0.653 and 0.965, respectively, with the Eckhardt's filter slightly outperforming the LH model across several metrics, including $NSE$, $KGE$ and $NRMSD$. This superiority extended to capturing peak timings and the shape of baseflow hydrographs, corroborated by stable and credible baseflow index values aligned with existing literature for comparable catchments. These results highlight the robustness of the Eckhardt filter and reinforce its suitability for baseflow separation in small, rural basins — particularly when $BFI_{max}$ can be calibrated to local conditions.

The introduction of an event-based calibration approach in this study marked a significant advancement in the field, demonstrating that baseflow separation accuracy could be enhanced by adjusting the $BFI_{max}$ and $Beta$ parameters, of Eckhardt's and LH filters, respectively, according to the magnitude of rainfall-runoff events. This approach yielded more precise baseflow estimations, as evidenced by improved performance indicators and a closer agreement with observed (DSi-derived) data. The findings revealed that the parameters $BFI_{max}$ and $Beta$ are not static but vary with event magnitude, highlighting the importance of the development of flexible calibration approaches for the Eckhardt's and LH filters. In particular, the Eckhardt's filter exhibited superior performance under event-based calibration, making it the most robust option. Optimal parameter values varied across event classes, with $BFI_{max}$ ranging from 0.809 for low-magnitude events to 0.576 for high-magnitude events, and $Beta$ ranging from 0.921 to 0.970, respectively. These results highlight that relying on a fixed, "one-size-fits-all" value — such as the commonly cited default of 0.25 for $BFI_{max}$ — can lead to significant underestimation of baseflow, especially in environments with dynamic hydrological responses.

In the presence of tracer data, event-based calibration of the Eckhardt's filter should be prioritized, as it substantially improves baseflow estimation across a range of flow conditions. Where tracer data are not available, practitioners are encouraged to identify catchment-specific $BFI_{max}$ values through literature review and site-based analysis. For catchments with similar characteristics to those described above, a general $BFI_{max}$ value of 0.653 is proposed as an appropriate starting point. This

value can be refined using established procedures, such as those described by Collischonn and Fan (2013), ensuring more realistic separation results even in data-scarce contexts.

Although this study focused on a single small catchment (1.2 km²), the findings are likely transferable to a variety of catchments, particularly those with pronounced rainfall variability and seasonal shifts in surface–groundwater interactions. Subtropical and semi-arid regions, where wet-season events generate high surface runoff and dry-season flows are groundwater-dominated, are especially well-suited for the application of event-specific RDF calibration.

Importantly, the methodology developed here is adaptable to both data-rich and data-limited environments. While DSi was used as a tracer in this study, other more accessible tracers — such as electrical conductivity — can serve a similar role in validating baseflow estimates. In the absence of any tracer data, the proposed general calibration framework remains valuable for guiding parameter selection and ensuring improved performance over default values.

To fully realize the potential of this approach, future research should pursue several strategic directions. First, event-based calibration should be applied across a broader range of catchments with diverse hydroclimatic conditions to test its generalizability. Second, including more events of varying magnitudes — from minor rainfall to extreme storms — will deepen our understanding of how calibration parameters respond under different hydrological conditions. Third, the development of systematic criteria for when to apply event-specific *versus* general calibration will aid practitioners in choosing the appropriate method for a given context. In addition, given the uneven seasonal distribution of events in the present study, we recommend that future research include a more seasonally balanced dataset to explore whether $BFI_{max}$ and *Beta* values also exhibit systematic seasonal variability. Understanding whether these filter parameters are influenced not only by event magnitude but also by broader seasonal hydrological controls could help refine calibration strategies and improve baseflow separation accuracy throughout the year.

Further opportunities lie in integrating this calibration approach with machine learning techniques to automate the adjustment of RDF parameters based on real-time hydrometeorological data. Such integration could improve operational baseflow separation by identifying patterns in the relationship between event characteristics and optimal parameter values. Additionally, exploring correlations between filter parameters and easily measurable catchment descriptors may enable regionalization schemes — extending the benefits of this methodology to ungauged or poorly monitored basins.

## 6. Appendices

**Table A1.** Error metrics comparison for observed *versus* simulated baseflows across 15 events, utilizing the event-based calibration of the **Eckhardt's filter** ($a = 0.952$, $BFI_{max}^{(low)} = 0.809$, $BFI_{max}^{(medium)} = 0.701$, $BFI_{max}^{(high)} = 0.576$). Acceptable model performance criteria: $NSE > 0.50$, $KGE > -0.41$, $NRMSD < 20\%$ and $PBias < +/- 25\%$

| Event - Magnitude | $Q_{max}$ (l.s⁻¹) | DSi samples | PBias (%) | NSE | KGE | NRMSD (%) | BF ratio |
|---|---|---|---|---|---|---|---|
| 1 – High | 3277.6 | 14 | 0.1**** | 0.88**** | 0.78**** | 11.7*** | 0.56 |
| 2 – Low | 56.0 | 7 | 25.7#### | -0.44 #### | 0.49 | 41.1## | 0.79 |
| 3 – Medium | 518.4 | 9 | -12.6* | 0.45** | 0.73* | 23.5* | 0.69 |
| 4 – Low | 43.2 | 4 | -6.5**** | -0.58**** | 0.52 | 47.7**** | 0.79 |
| 5 – Low | 35.3 | 6 | 0.7**** | -0.77*** | -0.56* | 45.4** | 0.78 |
| 6 – Low | 230.8 | 8 | 6.6**** | 0.72*** | 0.68 ### | 16.5** | 0.78 |
| 7 – Medium | 532.4 | 6 | 2.3**** | -0.63 # | 0.31 # | 43.6* | 0.62 |
| 8 – Medium | 445.3 | 10 | -3.9**** | -0.12* | 0.36* | 44.5 | 0.67 |
| 9 – Low | 125.1 | 3 | -24.5*** | 0.30**** | 0.75**** | 34.7*** | 0.78 |
| 10 – Medium | 493.0 | 13 | -0.4**** | 0.76 | 0.78 ## | 16.4 | 0.65 |
| 11 – Medium | 535.1 | 11 | 4.7 # | 0.31 | 0.65* | 26.7 | 0.69 |
| 12 – Low | 195.8 | 6 | -3.9**** | 0.43*** | 0.59** | 22.6** | 0.79 |
| 13 – Medium | 582.7 | 15 | -2.2**** | 0.43* | 0.70* | 18.7* | 0.69 |
| 14 – Medium | 998.0 | 6 | 0.7**** | 0.81** | 0.90** | 14.7* | 0.68 |
| 15 - Medium | 353.2 | 3 | 16.5 #### | 0.42 #### | 0.78 #### | 33.4#### | 0.68 |
| *All events combined* | | 121 | < 0.50 | 0.92*** | 0.91**** | 5.4** | 0.65 |

Changes relative to the general calibration: *5-20%; **20-35%; ***35-50%; ****>50% improvement. # 5-20%; ##20-35%; ###35-50%; ####>50% worsening. The absence of (*) or (#) indicates no significant change.

**Table A2.** Error metrics comparison for observed *versus* simulated baseflows across 15 events, utilizing the event-based calibration of the **LH filter** ($\beta^{(low)} = 0.921$, $\beta^{(medium)} = 0.957$, $\beta^{(high)} = 0.970$). Acceptable model performance criteria: $NSE > 0.50$, $KGE > -0.41$, $NRMSD < 20\%$ and $PBias < +/- 25\%$

| Event - Magnitude | $Q_{max}$ (l.s⁻¹) | DSi samples | PBias (%) | NSE | KGE | NRMSD (%) | BF ratio |
|---|---|---|---|---|---|---|---|
| 1 – High | 3277.6 | 14 | -0.3*** | 0.70** | 0.64** | 18.3* | 0.67 |
| 2 – Low | 56.0 | 7 | 26.9 #### | -0.87 | 0.35*** | 46.8 | 0.86 |
| 3 – Medium | 518.4 | 9 | -27.6** | -0.20* | 0.52* | 34.7* | 0.80 |
| 4 – Low | 43.2 | 4 | 10.0 #### | -1.38* | 0.26*** | 58.5 | 0.94 |
| 5 – Low | 35.3 | 6 | 4.4**** | -1.40** | -0.48 # | 52.9* | 0.81 |
| 6 – Low | 230.8 | 8 | 4.3 # | 0.50*** | 0.57 # # | 22.2** | 0.82 |
| 7 – Medium | 532.4 | 6 | -14.4** | -0.93* | 0.27 | 47.4* | 0.54 |
| 8 – Medium | 445.3 | 10 | -18.5** | -1.01* | -0.14* | 59.6* | 0.67 |
| 9 – Low | 125.1 | 3 | -28.5** | -0.32**** | 0.61**** | 47.8*** | 0.80 |
| 10 – Medium | 493.0 | 13 | -6.7**** | 0.63* | 0.63 # | 20.3 | 0.64 |
| 11 – Medium | 535.1 | 11 | 14.8 #### | -0.17 | 0.43* | 34.9 | 0.90 |

| | | | | | | |
|---|---|---|---|---|---|---|
| 12 – Low | 195.8 | 6 | -4.3**** | 0.23*** | 0.53** | 26.2** | 0.88 |
| 13 – Medium | 582.7 | 15 | 12.3 #### | -0.53 | 0.39 | 30.7 | 0.84 |
| 14 – Medium | 998.0 | 6 | -3.9**** | 0.83**** | 0.92**** | 14.1** | 0.77 |
| 15 - Medium | 353.2 | 3 | 41.9 # | -3.20 # | 0.37 # | 90.1 # | 0.87 |
| *All events combined* | | 121 | < 1.20 | 0.84** | 0.85*** | 7.8* | 0.76 |

Changes relative to the general calibration: *5-20%; **20-35%; ***35-50%; ****>50% improvement. # 5-20%; ##20-35%; ###35-50%; ####>50% worsening. The absence of (*) or (#) indicates no significant change.

## 7. Acknowledgements

Funding for this research was partially provided by the *Coordenação de Aperfeiçoamento de Pessoal de Nível Superior - Brazil (CAPES)* under Finance Code 001. We express our gratitude for the *Young Talent Grant* (reference number 88887.893273/2023-00), jointly supported by *CAPES* and the *Federal University of Santa Maria (UFSM)*, awarded to Dr. Fernanda Helfer. Additional appreciation goes to Brazilian agencies like *FAPERGS* and *CNPq,* whose financial support has been vital for sustaining ongoing research efforts at the Arvorezinha catchment. We also wish to recognize *Griffith University* (Australia) for enabling Dr. Fernanda Helfer's involvement in this Brazilian project through the *Griffith Academic Studies Program 2023-2024*. Finally, while AI tools assisted with linguistic polishing, we confirm that all concepts, text, and scientific contributions in this work are entirely human-generated.

## 8. Data Availability

The data that support the findings of this study are available from the corresponding author upon request.

## 9. Author contribution

**FH:** Conceptualization, Methodology, Software, Validation, Formal Analysis, Investigation, Resources, Data Curation, Writing – Original Draft, Writing – Review & Editing, Visualization, Project Administration, Funding Acquisition. **FB:** Conceptualization, Methodology, Software, Validation, Formal Analysis, Investigation, Data Curation, Visualization, Funding Acquisition. **CB:** Conceptualization, Methodology, Validation, Investigation, Resources, Data Curation. **DA**: Conceptualization, Methodology, Validation, Investigation, Resources, Supervision, Project Administration, Funding Acquisition. **JM:** Investigation, Resources, Supervision, Funding Acquisition. **RT:** Conceptualization, Methodology, Investigation, Resources, Project Administration. **NS**: Formal Analysis, Investigation, Visualization.

## 10. Competing interests

The authors declare that they have no known competing financial interests or personal relationships that could have appeared to influence the work reported in this paper.

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

## 12. Supplementary Material

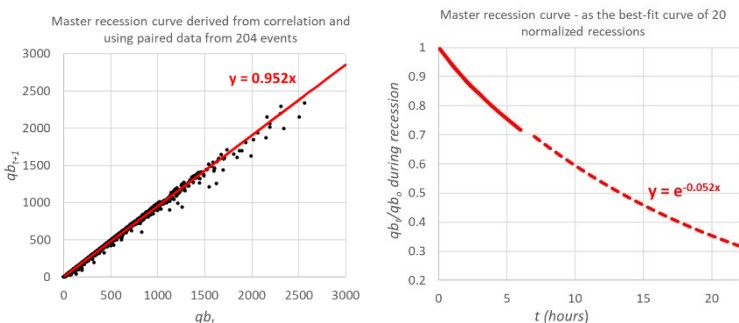


Figure S1. Master recession curve analysis for the Arvorezinha catchment derived from (left) correlation analysis of 204 recession events showing recession constant $a = 0.952$, and (right) best-fit exponential curve through 20 normalized recession hydrographs, with $k = -0.052$ ($a = 0.949$). The close agreement between methods validates the representativeness of the recession constant used in the baseflow separation filters.