# Peer review of "Enhanced Baseflow Separation in Rural Catchments: Event-Specific Calibration of Recursive Digital Filters with Tracer-Derived Data"

_EGUsphere, 2025_

## Author Response (AR1)

**Point-by-point reply to referees' comments**

Article: EGUSPHERE-2025-244

Anonymous Referee #1:

**General Comments:**

**This manuscript presents a well-executed and innovative study on baseflow separation in small rural catchments, focusing on the calibration of three Recursive Digital Filters (Eckhardt, Lyne and Hollick, and Chapman and Maxwell). The integration of dissolved silica as a tracer and the event-specific calibration approach are both novel and valuable contributions to the field. The writing is clear, and engaging, making the technical content accessible and enjoyable to read. The study's methodology is robust, and the results convincingly demonstrate the advantages of dynamic, event-based calibration-particularly for the Eckhardt filter, which outperforms the others in accuracy when parameters are tailored to rainfall event intensity.**

*Response:* *Thank you for your thorough and positive feedback on our study. We're pleased that our novel integration of dissolved silica and dynamic calibration approach has been recognised as a valuable contribution to the field.*

**Specific Comments:**

**While the manuscript is strong overall, there are areas where it could be further improved. The authors provide a thorough explanation of performance metrics such as NSE and RMSE, but this level of detail may be unnecessary for the target audience, who are likely already familiar with these standard evaluation tools. Streamlining these sections would help maintain the manuscript's focus and momentum.**

*Response:* *We thank the reviewer for their constructive feedback. We acknowledge the point that the target audience is likely familiar with the model performance metrics. So, to streamline the manuscript and maintain its momentum, we have condensed the descriptions of NSE, RMSE, and related metrics. We have reduced section 2.4 by 200 words. See revised section 2.4.*

**The presentation of results relies heavily on numerical values in the text. Incorporating more plots and figures would greatly enhance the reader's ability to interpret and appreciate the findings.**

*Response:* *We agree that more visual aids would enhance clarity and reader engagement. We have revised the results section (section 3), incorporating two additional multi-plot figures (Figures 7 and 8).*

**This is a timely contribution with clear novelty in its event-specific calibration strategy and use of chemical tracers. The manuscript is exceptionally well-written and methodologically sound. To maximize its impact, I recommend reducing the over-explanation of standard metrics and enhancing the presentation of results with more visual aids. These improvements would make the findings even more accessible and persuasive to a broad hydrology audience.**

*Response:* *Thank you again for the constructive feedback, which has improved the quality of our manuscript. As described in the above comments, we have incorporated both suggestions, reducing the over-explanation of performance metrics and including two new figures.*
* * *
Anonymous Referee #2

**General Comments**

**The manuscript describes an analysis in which three baseflow separation methods (specifically types of recursive digital filters) are tested against a silica-based estimate of baseflow for a small watershed in Brazil. The authors compare RDFs fit against the entire dataset as well as RDFs fit to events falling into one of three magnitude categories. The analysis shows that the Eckhardt filter tends to provide better results across events and that the use of categorized parameters can provide some improvement in estimating baseflow across a range of event types. The work**

presented is thorough and is explained with a commendable level of detail. The manuscript is organized nicely and is well written, making it easy to quickly understand what was done and the nature of the results. The authors provide a good explanation of the methods and the figures and tables support the interpretation and conclusions given.

*Response: Thank you for your positive and encouraging feedback. We are pleased to hear that you found the manuscript well-organized, clearly written, and methodologically sound, and we are grateful for your support in strengthening the manuscript.*

One criticism I might offer is that this work is based on and potentially limited in relevance to a single (and comparatively small) watershed. The manuscript would be more impactful if the value of the analysis is situated in a more generalized context - what about the results would be useful to a researcher or practitioner needing to understand or estimate baseflow somewhere else? I would suggest a potential expansion of the discussion and interpretation to address this broader context a bit more.

*Response: We appreciate the reviewer's insightful suggestion to strengthen the broader relevance of our findings. To address the limitation of focusing on a single, small watershed, we have substantially revised **Section 5 (Conclusions and Recommendations)** by adding five new paragraphs. These additions provide practical guidance for researchers and practitioners on how to adapt the event-specific calibration and tracer-based validation approach to a variety of catchment types, particularly under variable hydroclimatic conditions. We highlight the transferability of the recommended BFImax value (0.653) as a suitable starting point for catchments with similar characteristics and suggest alternative strategies for data-scarce regions, including the use of more widely available tracers such as electrical conductivity. Additionally, we expanded the final two paragraphs of **Section 5** to outline at least four key opportunities for future research. These include applying the method across diverse catchments and seasons, integrating machine learning for automated calibration, exploring regionalization techniques, and developing systematic protocols to guide the use of event-based versus general calibration. Together, these additions aim to position our study as a foundation for advancing practical, scalable baseflow separation strategies.*

**Specific and Technical Comments**

While the authors describe the physical characteristics of the watershed, a more specific explanation of the conceptual model of the watershed would be helpful in providing a basis for interpretation of the baseflow separation results. For example, it would be helpful to discuss what the Si data (or other previous work in the watershed) suggests about how baseflow contributes to streamflow. This can be used to provide context for the different methods and their interpretation too.

*Response: We appreciate the referee's suggestion to include a conceptual model of the Arvorezinha catchment to contextualize the baseflow separation results. We included a new paragraph in **Section 2.1** (last paragraph), describing our conceptual knowledge and understanding of the catchment's flow pathways, supported by previous studies carried out in the catchment. We hope these additions clarify the hydrological basis for our methods and enhance the interpretation of results for broader applications.*

What does the difference in performance from the tiered/categorized RDF parameter fits tell us about (potential) mechanisms in the watershed? This is an interesting result and I would appreciate a little more discussion on the topic.

*Response: We agree that this is a crucial point and have added two new paragraphs in **Section 3.5** to elaborate on the hydrological mechanisms underlying the variation in optimal BFImax and Beta parameters across different event magnitudes. Specifically, we explain that higher BFImax values observed during low-magnitude events reflect dominant groundwater contributions, as baseflow sustains most of the streamflow under these conditions. In contrast, lower BFImax values during high-magnitude events indicate increasing dilution of baseflow by rapid surface runoff. For the Beta parameter, we discuss how lower values during low-magnitude events suggest streamflow is largely governed by sustained groundwater discharge, while higher Beta values associated with high-magnitude events point to streamflow dominated by quickflow inputs and a reduced baseflow component.*

**The description indicates the area receives relatively consistent precipitation, perhaps more than many other places in the world. This would suggest fairly constant GW levels and perhaps less variation in "old" water. Yet the results show a lot of variability in shape and timing of baseflow. I'm curious how this is to be interpreted, and whether the authors view this tier-type approach as something than can be useful in disentangling seasonal variation in groundwater-surface water connection? Again, some additional discussion addressing this would be appreciated.**

*Response:* We respectfully disagree with the assumption that consistent annual precipitation necessarily results in stable groundwater levels or minimal variation in the "old water" (baseflow) contribution. While the Arvorezinha catchment exhibits relatively stable annual rainfall totals (average of ~1,938 mm), this precipitation is distributed unevenly throughout the year and often delivered in short, high-intensity events, particularly in the spring. Such intra-annual variability — combined with the catchment's steep topography, shallow soils, and fractured basalt geology — promotes rapid drainage and episodic recharge, leading to dynamic groundwater levels and highly variable baseflow contributions. This is evident in the diverse shapes and timings of the baseflow hydrographs across events (Figure 5), even within the same season. Therefore, consistent rainfall at the annual scale does not equate to uniform hydrological responses at the event scale, and variability in "old water" contributions remains an important feature of this system.

*Regarding seasonal variation, the event-based (tier-type) classification (low, medium, high magnitude) is driven by runoff response — not necessarily aligned with seasons. For example, low-magnitude events occurred in both spring (e.g., Event 5) and winter (e.g., Event 4), suggesting that hydrological behaviour, not season, governed parameter variation. So, while the event-based calibration successfully reveals variability in BFImax and Beta as a function of event magnitude, it does not support inference about seasonal variability in these parameters. Drawing seasonal conclusions would require: A larger number of events, more balanced seasonal coverage (including summer), and stratified analysis of events by both season and magnitude. We have added this as a recommendation for further research (see the second-to-last paragraph of* **Section 5 – Conclusions and Recommendations***).*

**Much of the information shown in error metrics tables is repetitive between the two sections of analysis and results. I'd suggest condensing some of this and/or moving some of it to an appendix/supplement.**

*Response:* We appreciate your suggestion, and we have moved Tables 9 and 10 (from the original manuscript) to the Appendices (now Appendices A1 and A2). This suggestion aligns with the first referee's comment about improving the presentation of results by using more visuals than tables, so we added two new figures in the results section.

**Table 1 (page 8) - Units for Qmax and Qmin are different. Check flow units for consistency.**

*Response:* Thank you for your suggestion. To maintain consistency with the units for Qmax, we have expressed Qmin in m3/s.

**Figure 2 and Figure 5 - I appreciate the information density on these plots. However, the line widths on the RDF results (colored lines) came through rather faintly on the PDF I reviewed, making it difficult to readily differentiate among the results. Additionally the figure seemed a bit blurry. To the extent that the line widths and resolution could be adjusted, these figures will be much more effective.**

*Response:* We have increased the resolution of Figures 2 and 5.

**Line 558: "events 2,9, and 9 showed..." - I suspect the second "9" was in error here.**

*Response:* We have fixed this typo.

---

## Author Response (AR2)

**Point-by-point reply to referees' comments**

Article: EGUSPHERE-2025-244

Anonymous Referee #2:

**Report #2**

Submitted on 15 Jul 2025
Anonymous referee #2

**Anonymous during peer-review: Yes** No
**Anonymous in acknowledgements of published article: Yes** No

**Checklist for reviewers**

**1) Scientific Significance**
Does the manuscript represent a substantial contribution to scientific progress within the scope of this journal (substantial new concepts, ideas, methods, or data)?

Excellent **Good** Fair Poor

**2) Scientific Quality**
Are the scientific approach and applied methods valid? Are the results discussed in an appropriate and balanced way (consideration of related work, including appropriate references)?

**Excellent** Good Fair Poor

**3) Presentation Quality**
Are the scientific results and conclusions presented in a clear, concise, and well structured way (number and quality of figures/tables, appropriate use of English language)?

**Excellent** Good Fair Poor

**For final publication, the manuscript should be**

**accepted as is**
accepted subject to **technical corrections**
accepted subject to **minor revisions**
reconsidered after **major revisions**
rejected

**Were a revised manuscript to be sent for another round of reviews:**
**I would be willing to review the revised manuscript.**
I would not be willing to review the revised manuscript.

**Suggestions for revision or reasons for rejection**
(visible to the public if the article is accepted and published)
I thank the authors for their careful consideration of my questions, comments, and suggestions on the previous version of the manuscript. I especially appreciate the additional explanation provided in several parts of the paper to clarify interpretation of the analysis. My questions have been addressed sufficiently - I can now recommend the manuscript be accepted for publication.

**Suggestions for revisions or reasons for rejection:**

**I thank the authors for their careful consideration of my questions, comments, and suggestions on the previous version of the manuscript. I especially appreciate the additional explanation provided in several parts of the paper to clarify interpretation of the analysis. My questions have been addressed sufficiently - I can now recommend the manuscript be accepted for publication.**

*Response: Thank you for your thorough and positive feedback on our study and confirming that the questions have been addressed sufficiently.*

Referee #3 – Marcus Gomes Jr - marcusnobrega.engcivil@gmail.com

**Report #1**

Submitted on 09 Jul 2025
Referee #3: Marcus Gomes Jr., marcusnobrega@gmail.com

**Anonymous during peer-review:** Yes **No**
**Anonymous in acknowledgements of published article:** Yes **No**

**Checklist for reviewers**

| | |
|---|---|
| **1) Scientific Significance**
Does the manuscript represent a substantial contribution to scientific progress within the scope of this journal (substantial new concepts, ideas, methods, or data)? | **Excellent** Good Fair Poor |
| **2) Scientific Quality**
Are the scientific approach and applied methods valid? Are the results discussed in an appropriate and balanced way (consideration of related work, including appropriate references)? | **Excellent** Good Fair Poor |
| **3) Presentation Quality**
Are the scientific results and conclusions presented in a clear, concise, and well structured way (number and quality of figures/tables, appropriate use of English language)? | **Excellent** Good Fair Poor |

**For final publication, the manuscript should be**

accepted as is

accepted subject to **technical corrections**

**accepted subject to minor revisions**

reconsidered after **major revisions**

rejected

Were a revised manuscript to be sent for another round of reviews:

**I would be willing to review the revised manuscript.**

I would not be willing to review the revised manuscript.

**Suggestions for revision or reasons for rejection**
* * *
**Suggestions for revisions or reasons for rejection:**

**Thank you for submitting the paper. This article contrasts the idea of time-invariant baseflow parameters and proposes an event-based calibration approach that enhances the performance of baseflow estimation in a small catchment. The article is clear, all sections are well written, and the article has clear merits, in my opinion. I do, however, have a few comments and suggestions listed below:**

**Suggestion: Please specify why the authors chose these 15 events in Table 1. Give the rationale for why they were selected.**

*Response: We thank the reviewer for this comment, which highlights the need for clarity regarding the selection of the 15 events presented in Table 1. The events were selected to represent a diverse range of hydrological conditions (hydrological variability) in the studied small rural catchment, ensuring robust calibration and validation of the RDFs. Moreover, the selected events had high-quality tracer-derived and streamflow data (DSi) available, which were critical for accurately distinguishing baseflow from quickflow components. Events with missing data or poor-quality measurements were excluded to maintain the integrity of the analysis. To address the reviewer's comment, we have added a paragraph in Section 2.1 Study site and data collection to clarify the selection criteria for the 15 events.*

**Suggestion: Please justify the rationale behind adopting 2 hours after the peak to determine the recession period. The literature includes different approaches for this—some use rainfall data, others use topographic information to determine catchment response times, etc. Did the authors perform a master recession curve analysis as well? If so, that would be interesting to include, at least as supplementary information.**

*Response: We appreciate the reviewer's important question about our recession period determination. The 2-hour threshold was selected based on several catchment-specific considerations:*

- *The Arvorezinha catchment exhibits a rapid hydrological response due to its size and topography, as described in 2.1 Study Site. Analysis of multiple observed hydrographs showed that quickflow contributions typically decline significantly within 1-2 hours after the peak, with baseflow dominating thereafter.*
- *Visual and statistical inspection of various hydrographs observed at this catchment, indicated a transition to a smoother, exponential decay pattern approximately 2 hours post-peak, suggesting baseflow dominance. This was confirmed by examining the rate of streamflow decline across multiple events.*
- *The 2-hour threshold provided a standardized starting point for analyzing recession behavior across 204 hydrographs, facilitating the estimation of the recession constant.*
- *Rainfall-based approaches were less suitable due to variability in rainfall cessation timing, which could introduce inconsistencies in a flashy catchment.*
- *Topographic methods were not adopted because event-specific factors (e.g., antecedent moisture) dominate the catchment's response, better captured through hydrograph analysis.*

*How we derived the recession constant: The recession constant in our study (0.952 or ≈ 0.0492 $h^{-1}$) was derived using 204 observed hydrographs, yielding over 17,000 pairs of consecutive streamflow values (q(t),q(t+1)). Recession periods were identified starting 2 hours post-peak to approximate baseflow dominance. All q(t) and q(t+1) pairs were compiled into two Excel columns, and a scatter plot of q(t+1) versus q(t) was created. A linear trendline, forced through the origin, was fitted, yielding a slope of 0.952, This approach assumes a consistent exponential decay model across all recession periods, capturing the catchment's average recession behavior. It is therefore, an MRC approach based on correlation between q(t) and q(t+1) for all observations available.*

*To address the reviewer's suggestion, we conducted another MRC analysis using a sub-set of 20 randomly-selected hydrographs to validate our correlation-based approach described above. In this method, streamflow data were normalized (qt/q0) starting 2 hours post-peak (i.e. q0 is the first flow at the beginning of the recession phase for each hydrograph), and the average normalized streamflow was calculated for each time step to form a composite recession curve representing the 20 events. Linear regression of the log-transformed data (ln(qt/q0)=t*ln(a)) yielded an MRC recession constant of a=0.949 (≈ 0.053 $h^{-1}$). This value is close to the value found using the approach that we described in the paper, with all recession periods combined. This also confirms the robustness of the 2-hour criterion. The slight difference reflects the MRC's smoothing of event-specific variability through normalization and averaging, whereas our method treats all q(t),q(t+1) pairs equally, potentially amplifying short-term fluctuations.*

*To reflect this additional validation, we have substantially revised Section 2.2 (methodology for recession constant estimation) to include detailed description of both the original paired-data approach and the MRC analysis, and Section 3.1 (recession constant results) to present the comparative findings. The plots for both methodologies are included in the Supplementary Material section (Figure S1), as suggested by the reviewer.*

**Suggestion:** **Please specify which optimization algorithm was used to find the near-optimal or optimal BFImax and Betamax. In which programming language was it developed?**

*Response: We thank the reviewer for requesting clarification on our optimization methodology. The BFImax and Beta parameter optimization was performed using the bisection method implemented in MATLAB. This robust numerical root-finding algorithm iteratively narrows the search interval [0.001, 0.999] for BFImax and [0.900, 0.999] for Beta by evaluating PBias at successive midpoints until convergence within a precision tolerance of 0.001. The bisection method was selected for its guaranteed global convergence, robustness against local optima, and computational transparency in demonstrating parameter sensitivity across the full feasible range. Complete algorithmic details have been added to Section 2.3 of the revised manuscript.*

**Suggestion:** **Please provide the rationale for choosing PBIAS as the objective function.**

*Response: We thank the reviewer for this important methodological question. PBias was selected as the objective function because it directly quantifies systematic bias in baseflow volume estimates, has a clear optimal target of zero that is well-suited for our bisection method optimization, provides dimensionless percentage errors enabling comparison across events of different magnitudes, and is widely used in hydrological calibration studies, ensuring consistency with established literature. While PBias served as the primary optimization criterion, we also evaluated model performance using complementary metrics (NSE,*

*KGE, NRMSD) to provide comprehensive assessment. The rationale for this choice has been added to Section 2.3 of the revised manuscript.*

**Suggestion: Fix the typo in line 455 (BFImax).**

*Response: We have fixed this typo.*

**Suggestion: In the conclusions, please specify the overall catchment characteristics when providing the optimal values of Beta and BFImax. The authors mention this later, but it would be better to include it at the beginning.**

*Response: We thank the reviewer for this excellent suggestion to improve clarity and applicability. We have revised the conclusions section to specify the catchment characteristics (small area, steep slopes, rapid baseflow recession, fractured basalt geology, mixed soil types) upfront when presenting the optimal BFImax and Beta values of 0.653 and 0.965, respectively. This modification makes it immediately clear to readers under what hydrological and geological conditions these parameter values are applicable, enhancing the practical utility of our recommendations for researchers working in similar environments.*

**Suggestion: Do the values of BFImax and Beta, when calibrated using event-based formulations, represent a characteristic response of the catchment, or are they more reflective of uncertainty? Please clarify. The recession constant is generally considered a characteristic response of the catchment and is typically assumed to be time-invariant, but apparently, Beta and BFImax are not.**

*Response: We appreciate this insightful question. We agree that the recession constant reflects a time-invariant, physical characteristic of the catchment — primarily governed by geomorphology, topography, and aquifer properties — and we have treated it as such in our study.*

*In contrast, BFImax (Eckhardt's filter) and Beta (Lyne & Hollick filter) are calibration parameters that are more sensitive to hydrological conditions during individual events, particularly the relative contributions of baseflow and quickflow. Therefore, the variations in BFImax and Beta under different event magnitudes should not be interpreted as uncertainty in the methodology, but rather as an expression of dynamic catchment response under varying hydroclimatic forcings. This interpretation aligns with previous findings (e.g., Zhang et al., 2013; Okello et al., 2018), which demonstrated seasonal and event-dependent variation in these parameters.*

*Our event-based calibration shows systematic trends: for example, BFImax decreases with increasing event magnitude, reflecting the dilution of baseflow by quickflow during intense storms. Similarly, Beta values increase with event magnitude, as fast quickflow components dominate and slow responses are proportionally reduced. These variations are not arbitrary or uncertain — they are hydrologically meaningful and reflect how the catchment's flow partitioning behavior changes across events, despite having a stable recession constant.*

*We now clarify this interpretation in Section 3.5 (last paragraph) of the revised manuscript.*

**Suggestion: In the original publication by Eckhardt (2005), several criticisms of BFImax estimation were discussed, since it is not directly measurable like the "a" parameter. Given this, the authors developed an event-based framework to estimate BFImax, varying across all events. The coefficients of variation from the Eckhardt filter were relatively low compared to other methods, indicating some convergence of BFImax values. This may suggest that BFImax is a characteristic response of the catchment. In contrast, such convergence was not clearly observed using methods like Lyne-Hollick (LH). Please discuss this further.**

*Response: We appreciate the reviewer's insights. In the general calibration, the BFImax is fixed for all events. So the resulting baseflow ratios (BF ratio) are very similar across events, because the same parameter is applied universally. Even if some events are slightly under or overestimated, the output is numerically smooth. Therefore, both mean and standard deviation are small, and CV is low (0.04). In the event-based calibration, we used three different BFImax values — 0.809 (low), 0.701 (medium), 0.576 (high magnitude). This means low events yield high baseflow ratios (BFImax = 0.809), high events yield low baseflow ratios (BFImax = 0.576), and therefore this introduced a broader spread in BF ratios. This naturally increases the standard deviation of baseflow ratios, but the mean remains similar (around 0.65), leading to an increased CV (0.09). So, the increased CV is expected — it reflects that we're intentionally tailoring BFImax to reflect different hydrological conditions across events, resulting in more variability in the outputs. This is not a flaw — it shows*

we are capturing real physical variability that the general model smoothed over. The physical interpretation for this result is that event-based calibration reflects hydrologic reality: In low-magnitude events, the catchment is dry and streamflow is mostly groundwater → high baseflow contribution → higher BFImax. In high-magnitude events, overland flow dominates → low baseflow contribution → lower BFImax. So the increase in CV under event-based calibration mirrors real differences in baseflow generation processes.

The Lyne-Hollick (LH) filter, on the other hand, has only one parameter, Beta, and it operates very differently from Eckhardt's filter: LH separates quickflow directly, and baseflow is just the remainder. It's more numerically damped and less sensitive to Beta than Eckhardt's filter is to BFImax. Even though we changed Beta for different event classes (0.921, 0.957, 0.970), the resulting baseflow ratios did not change much, because the structure of the LH filter is inherently less responsive to Beta adjustments; we already used a relatively high Beta (around the recession constant), so small changes had diminishing returns; and therefore, standard deviation of baseflow ratios didn't increase much, so the CV remained stable (~0.15 to 0.14). Physically, this suggests that LH's filtering behavior does not capture hydrologic variability across event magnitudes as well as the Eckhardt's filter does. In other words, Eckhardt's filter is more sensitive to parameter changes, and therefore better at reflecting true hydrological differences (at the cost of increased CV). LH is more rigid, so even when calibrated dynamically, it doesn't translate into much variability in baseflow ratios — hence the CV doesn't change much.

We have clarified this point in the revised manuscript (Section 3.6).

**Since the parameters were only calibrated and no validation test was performed—due to the assumption of event-specific, time-varying catchment responses—the results might be overfitted. I would suggest quickly splitting the time series into two parts, calibrating the optimal BFImax on one set of events (event-based), computing the mean parameter, and evaluating the performance on the remaining events, if this is not too time-consuming. Would a time-invariant estimate of BFImax perform better than a time-varying parameter estimate using the LH method, for instance? If so, this would support better performance of the Eckhardt filter compared to LH. Please feel free to argue this point. Alternatively, since the authors used only 15 events out of potentially hundreds, a quick validation test would help increase confidence in the results.**

Response: We appreciate the reviewer's suggestion for additional validation using independent events, which would indeed provide valuable insights into the generalizability of our event-based calibration approach. However, we respectfully note that implementing this validation framework presents significant practical constraints that prevent its inclusion in the current study.

Data availability constraints: While we have an extensive database of over 200 hydrographs for the Arvorezinha catchment, dissolved silica measurements were only collected during the 15 events presented in this study. Expanding silica monitoring to additional events would require substantial field campaigns and analytical resources that extend beyond the current study's scope.

Methodological foundation: Our event-based calibration approach was developed based on the established understanding that baseflow indices vary with hydrological conditions, as demonstrated in the literature (Minea, 2017; Okello et al., 2018; Zhang et al., 2013). Therefore, BFImax - which directly controls the maximum baseflow contribution - should logically vary with event magnitude. Our study confirms this theoretical expectation through empirical evidence, rather than challenging an established paradigm.

Acknowledgment of limitations: We acknowledge that the current calibration dataset is limited and that overfitting concerns are valid. However, this represents a foundational study that establishes the proof-of-concept for event-specific calibration. As additional tracer data becomes available through future monitoring campaigns, the robustness and transferability of our calibrated parameters will be systematically evaluated and refined.

Future research direction: The validation framework suggested by the reviewer represents an excellent direction for future research and will be prioritized as expanded datasets become available.

This has been articulated in the second-last paragraph of the conclusion section.